# Solving dynamic multi-objective problems with a new prediction-based optimization algorithm

**Qingyang Zhang** **[1]\*, Shouyong Jiang[2], Shengxiang Yang[3], Hui Song[1]**

**1** School of Computer Science and Technology, Jiangsu Normal University, Xuzhou, CO, China, **2** School of Computer Science, University of Lincoln, Lincoln, CO, United Kingdom, **3** School of Computer Science and Informatics, De Montfort University, Leicester, United Kingdom

\* sweqyian@126.com

**Data Availability Statement:** All relevant data are within the manuscript and its Supporting information files.

**Funding:** This work is supported by the National Natural Science Foundation of China under Grants

## Abstract

This paper proposes a new dynamic multi-objective optimization algorithm by integrating a new fitting-based prediction (FBP) mechanism with regularity model-based multi-objective estimation of distribution algorithm (RM-MEDA) for multi-objective optimization in changing environments. The prediction-based reaction mechanism aims to generate high-quality population when changes occur, which includes three subpopulations for tracking the moving Pareto-optimal set effectively. The first subpopulation is created by a simple linear prediction model with two different stepsizes. The second subpopulation consists of some new sampling individuals generated by the fitting-based prediction strategy. The third subpopulation is created by employing a recent sampling strategy, generating some effective search individuals for improving population convergence and diversity. Experimental results on a set of benchmark functions with a variety of different dynamic characteristics and difficulties illustrate that the proposed algorithm has competitive effectiveness compared with some state-of-the-art algorithms.

## 1 Introduction

The progress of optimizing multiple mutually conflicting objectives simultaneously and obtaining a set of tradeoff solutions is regarded as Multi-objective optimization problems (MOPs) [1], which involves different fields, including controller design [2], weapon selection [3] and machine learning [4]. Simultaneously, various multiobjective optimization algorithms have been proposed for solving MOPs successfully. Considering a minimization multiobjective optimization problem as follows,

$$\min_{x \in \Omega} F(x) = (f_1(x), f_2(x), \ldots, f_m(x))^T \tag{1}$$

where $\Omega = \prod_{i=1}^{D} [L_i, U_i] \subset R^D$ is the feasible area of the decision space, and $F$ consists of $m$ time-varying objective functions. $x = (x_1, x_2, \ldots, x_D)$ defines the decision vector involving $D$ variables, $L_i$ and $U_i$ represent the lower and upper bounds of the $i$th variable $x_i$, respectively.

62006103 and 61872168, in part by the Jiangsu national science research of high education under Grand 20KJB110021.

**Competing interests:** The authors declare that they have no known competing financial interests or personal relationships that could have appeared to influence the work reported in this paper.

For two given decision vectors $x$ and $y$, if $\forall j \in [1, m], f_i(x) \leq f_i(y)$ and $\exists l \in [1, m] f_l(x) < f_l(y)$, then, $x$ dominates $y$ regarded as $x \prec y$. If a vector $x^*$ can dominate any other solutions, $x^*$ is defined as Pareto optimal solution.

However, recent years, there exist an increasing number of multi-objective optimization problems recognised in various fields, such as scheduling [5, 6], planning [7, 8], resources allocation [9, 10], constrained optimization [11], and machine learning [12], needed to be solved in dynamic or uncertainties environment, which are named dynamic multi-objective optimization problems (DMOPs). The main characteristic of this kind of problem is that the constraints, the Pareto optimal set (POS) or Pareto-optimal front (POF), and the relevant control parameters can change dynamically, which brings great challenges to optimization algorithms. It has attracted a growing attention for exploring efficient optimization algorithms and obtaining high quality optimal solution sets. Although there may exist different classes of dynamic optimization problems, according to [1], this paper considers the following mathematical model of DMOPs.

$$\min_{x \in \Omega} F(x, t) = (f_1(x, t), f_2(x, t), \ldots, f_m(x, t))^T \qquad (2)$$

where $t$ is the time instant of the problem.

Compared with MOPs, dynamic dynamic multi-objective optimization problems have two important features: multiobjectivity and dynamism. It is generally known that multiobjectivity usually involves multiple conflicting objectives, which means the optimal solution of the problem will no longer be a single optimal value, but an optimal solution set containing tradeoff solutions. Dynamism in constraints and/or parameters causes the change of POF or POS and poses big difficulties to evolutionary algorithms. DMOPs are challenging due to the dynamic nature. They can be divided into a sequence of MOPs over the course of time. That is, the optimization goal is to obtain a sequence of approximations to the moving POS/POF.

## 2 Related work

In recent years, much effort has been devoted to designing efficient and effective dynamic multi-objective evolutionary algorithms (DMOEAs). A widely used framework of DMOEAs in literature can be described as Algorithm 1. As shown in this framework, the whole procedure of solving DMOPs contains two main components: change detection and multi-objective algorithms including MOEAs and DMOEAs.

**Algorithm 1** The basic framework of DMOEA

```
1: Initialize time instance t = 1;
2: Generate an initial population Pop_t;
3: While the termination criterion is not satisfied
4: Change Detection
5:   If change is not detected, evolve population using MOEA;
6:   Otherwise, evolve population using DMOEA;
7: Return 3.
```

### 2.1 Change detection

As a significant component of DMOEAs, change detection is responsible for determining whether the environment has changed and in turn whether to implement a reaction mechanism. The existing dynamic extraction methods contain two categories: re-evaluating solutions [13–15] and checking population statistical information [16]. The former is more widely used in many algorithms because it is simple and easy to implement, but it is likely sensitive to

noise. In contrast, the latter is robust to noise, but it needs some additional parameters. Each method has its advantages and limitations for different DMOPs.

## 2.2 Multi-objective optimization algorithms

Apart from the dynamic reaction mechanism, MOEAs are significant components of solvers for DMOPs, since DMOPs can be regards as a sequence of MOPs. That is, any MOEAs can be directly used to evolve the population during the (short) period of any static environments.

As one of the most attractive and popular areas in intelligent computing field, the existing Multi-objective optimization algorithms can be classified into three categories as follows. The first class is Pareto ranking-based algorithms, which are designed based on the dominated relationships among population individuals. Some representative algorithms include the nondominated sorting genetic algorithm II (NSGA-II) [17], and strength Pareto evolutionary algorithm (SPEA2) [18]. Besides that, some classic and recent proposed efficient swarm intelligence algorithms inspired by different nature behaviors have also used to solve MOPs, such as multi-obejctve particle swarm optimization (MOPSO) [19], Multi-Objective Grasshopper Optimisation Algorithm (MOGOA) [20], Multi-Objective Multi-Verse Optimizer (MOMVO) [21], Multi-Objective Ant Lion Optimizer [22], and Multi-objective Salp Swarm Algorithm (MSSA) [23], and so on. Although the non-dominant ranking strategy can well screen out excellent individuals, it also produces marginal individuals, which generate negative effect on the whole optimization process. These algorithms can obtain good local optimal solutions, but it is difficult to achieve ideal global optimal solutions.

The second class is indicator-Based algorithms, which are designed based on the performance indicators. The hypervolume [24], the epsilon indicator and the R2 one are the most utilized for proposed various algorithms, such as, indicator-based EA (IBEA) [25], S-metric selection EMO algorithm [26], R2 EMO algorithm (R2EMOA) [27], and approximation-guided EMO (AGE) [28]. The last class is decomposition-Based algorithms, which aim to decompose the MOP into some optimization sub-problems and solve them simultaneously. The most used algorithms are NSGA-III [29, 30], and MOEA based on decomposition (MOEA/D) [31, 32]. Although this kind of algorithm is efficient, the division of sub-problems depends on the weight deeply.

## 2.3 Dynamic multi-objective optimization algorithms

Depending on the frequency or severity of change, changes may present various challenges, such as the finite computational time or resources to overcome the change, time-varying feasible region and constraint conditions. Therefore, effective and efficient dynamic multi-objective optimization algorithms are indeed important. Diversity and convergence are two important aspects in designing high-quality optimization methods, since the former aims to prevent the search from local optima whereas the latter helps algorithms to find promising solutions rapidly. Designing an effective strategy that is able to balance the diversity and convergence is one of the key topics in DMOPs. Existing Dynamic multi-objective optimization algorithms can be divided into four categories: diversity based algorithms, memory based algorithms, multi-population based algorithms, and prediction based algorithms.

The main purpose of diversity based algorithms is to maintain the search population diversity for avoiding local optima when a change is detected. Recently, a increasing number of diversity maintenance methods have been proposed. A general framework proposed by Li [33] maintains the diversity by utilizing hierarchical linkage clustering, which is able to generate subpopulations with good diversity while avoiding overlapping. Query-based strategy proposed by Chang et al [34] increases the population diversity by providing a guidance to

particles. Immigration-based strategy aims to prevent local optima and achieve better search ability, such as hybrid immigration [35], memory-based immigration [36] and elitism-based immigration [37]. Besides that, hyper-mutation has been employed to combine with the non-dominated sorting genetic algorithm II (NSGAII) [38] to create two different dynamic versions for DMOPs.

The main idea of memory based algorithms is to record some historical information, which can be reused to accelerate the convergence of algorithms whenever a change occurs. Branke [39] suggests that the best individuals in previous change environments were stored in an archive firstly, and used to replace some members of the existing population. Goh [40] proposed a strategy that employs an new population to replace the out-of-date archived members, which integrates competitive and cooperative mechanisms for DMOPs. In [41], memory, local search and random techniques are integrated, and an adaptive hybrid population management strategy is proposed by authors. Jiang and Yang [42] used a steady-state manner to respond to changes. These kinds of algorithms performs well on problems with periodical changing feature.

The main idea of multi-population based algorithms is that multiple subpopulations can be advantageous at maintaining diversity. In [43], a self-organizing scouts method is proposed by dividing the search population into two subpopulations, which are used to search in feasible regions. Li [44] combined an island model with particle swarm optimization for dealing with dynamic vehicle routing problems. Yang [45] employs hierarchical clustering to divide the population into several subpopulations of different sizes for effective diversity maintenance [46].

Prediction based algorithms aims to predict a possible POF/POS locations of new environments based on the solutions in previous environments. These algorithms are much popular in DMOEAs, since prediction-based mechanisms could help tracking the moving POS/POF if solutions in new environments are well predicted. Muruganantham [47] proposed a DMOEA by combining Kalman filter with evolutionary methods for solving DMOPs. The multimodal prediction approach proposed by Rong [48] refers to generate an effective initial population for the subsequent evolution. Population Prediction Strategy (PPS) [49] proposed by Zhou *et al.* is used to predict the manifold of the whole search population by using the univariate auto-regression (AR) model. Besides that, many other prediction approach have been proposed in different ways, such as multi-directions [50], knee points [51], center points [52], and boundary points [53].

Most of the existing DMOEAs have been proposed, showing promising performance in various applications. However, they neglect properties of decision variables, which is an important part of discovering high-quality search individuals. Simultaneously, according to [54, 55], curve fitting technique is a classic and popular technique, which can reflect the distribution relationship between variables to a certain extent and predict possible regions or directions. Motivated by this, this paper proposed a novel method for predicting a high-quality population based on the distribution and classification characteristics of variables after a change is detected. The proposed algorithm contains three different parts, firstly, a simple linear prediction strategy with two different stepsizes is designed to predict non-dominated solutions based on the information of previous environments. The second strategy is proposed by integrating fitting-based strategy for generating new members and improving the quality of population based on the probability distribution of variables. The last strategy aims to generate well-distributed individuals based on the classification features of decision variables. Numerical results on 14 benchmark functions show that the proposed algorithm performs well on tracking time-varying POF or POS.

The following summarizes the organization of this work. Section 2 presents the related work. The proposed algorithm is provided in Section3. In Section 4, the performance of the proposed technique is validated and analyzed on a comprehensive set of benchmark functions. Section 5 gives further discussion about the proposed algorithm. Section 6 concludes the paper.

## 3 Proposed DMOEA

This section mainly provides the main content of the proposed algorithm in detail. Like other predicted algorithms, our hypothesis is that there is sort of similarity between two consecutive changes. As obtained from the basic framework of the proposed algorithm listed in Algorithm 2, the main idea combines RM-MEDA with a new prediction-based dynamic reaction mechanism, which has three different strategies for predicting a new high-quality search population that tracks the new POS/POF efficiently and effectively.

**Algorithm 2** The overall framework of the proposed algorithm

```
1: Initialize parameter settings.
2: Initialize and evaluate population (PopGen) and set Gen = 1.
3: If the stop condition is not satisfied.
4: If change detected, go to step 5; otherwise, go to step 10.
5: Generate the first subpopulation (SubPop1) using a linear predic-
tion model.
6: Generate the second subpopulation (SubPop2) based on new fitting-
based strategy.
7: Generate the third subpopulation (SubPop3) by recent proposed sam-
pling strategy [56].
8: Merge these subpopulations MixPop = SubPop1∪SubPop2 ∪ SubPop3.
9: Obtain a population of Popsize by non-dominated sorting the merged
population.
10: Optimize population using RM-MEDA.
11: Gen = Gen + 1, return to 3.
```

### 3.1 Linear prediction model

This subsection mainly employs a simple linear prediction model with two different stepsizes for predicting non-dominated set. From statistical point of view, the geometric center is an important characteristic and can be used to represent the changing trend of population to some extent. Here, we compute the moving direction of the center points of the last two consecutive populations and use it to predict the position of the non-dominated members of current population in the new environment.

Suppose that $Pc_t$ is the centroid of population ($Pop_t$) and $Pos_t$ is the non-dominated sets of $Pop_t$ at the time $t$. Then, the $pc_t$ can be calculated as follows.

$$Pc_t = \frac{\sum_{x_t \in Pop_t} x_t}{|Pop_t|} \tag{3}$$

where $|Pop_t|$ is the population size, $x_t = (x_t^1, x_t^2, \ldots, x_t^D)$ defines the decision vector of a solution at time $t$. Then, the moving direction ($dir_t$) of center points at time $t$ can be calculated by

$$dir_t = pc_t - pc_{t-1}. \tag{4}$$

Then, the new position of members in $Pos_t$ at time $t + 1$ can be obtained by $dir_t$ and $Pos_t$ according to the following formula:

$$Pos_{t+1} = Pos_t + dir_t \times step \tag{5}$$

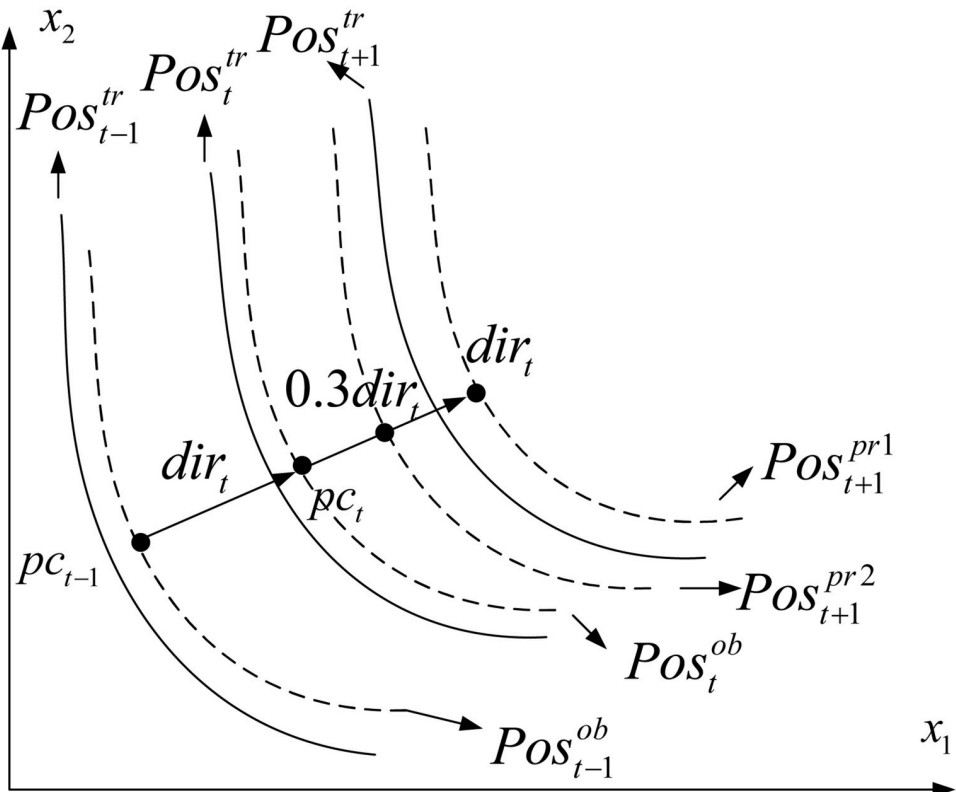

**Fig 1. Illustration of linear prediction model.**

where *step* refers to the moving stepsize along the moving direction of $dir_t$. Here, two different values of *step* (i.e., 0.3 and 1.0,) are used, representing a small and large movement of $Pos_t$, respectively. Fig 1 illustrates the prediction process.

As shown in Fig 1, $pc_t$ and $pc_{t-1}$ (black points) are utilized to obtain $dir_t$. $Pos_t$ moves to three different regions described by $Pos_{t+1}^{pr1}$ and $Pos_{t+1}^{pr2}$ using the suggested *step* values. A combination of these two solution predictions is more likely to approximate the true POS of population ($Pos_{t+1}^{tr}$) at time $t+1$. Algorithm 3 provides the implementation of this prediction strategy.

Two questions may arise here, on the one hand, the motivation about the two-step prediction strategy to produce good individuals. In the ideal environment, The widely used one-step strategy assumes the change between two continuous times is same to some extent. This proves effective in various algorithms and we would also like to keep it in our algorithm. However, as suggested by [38], sometimes a small variation to the population can be very effective. This inspires us that a smaller stepsize than the previous stepsize would be helpful in creating population individuals for environments that do not change significantly. That is, a smaller moving step may ensure that the predicted solution is much closer to the new POS after a change. As a result, this work attempts to design a two-step prediction strategy for DMOPs.

On the other hand, how to determine stepsize parameters is a major issue. The proposed strategy employs two stepsize values (0.3 and 1.0), which represents two different moving levels (small and normal). There are two reasons for this setting. First, *step* = 1 for the normal level is set according to fuzzy systems [57], which means that the change is similar to the previous change (normal changes). Second, the stepsize *step* = 0.3 for the small level should be

smaller than that for the normal level. The stepsize setting is chosen not only for simplicity but also by sensitivity analysis as will be detailed in in Section 4.

**Algorithm 3** Linear prediction model

```
1: Retrieve the populations Pop_t and Pop_{t-1} at time t and t - 1,
respectively;
2: Calculate the population centers according to Eq (2);
3: Predict moving direction according to Eq (3);
4: Generate three subpopulations Pos_{t+1}^{pr1} and Pos_{t+1}^{pr2} using Eq (4) with dif-
ferent step values;
5: Save the subpopulations to SubPop_1.
```

### 3.2 Curve fitting-based strategy

This subsection proposes a curve fitting-based strategy for generating high-quality search individuals based on the distribution relationship of variables. As suggested in [56], the variables can be classified into two parts: principal and non-principal parts. We believe the correlation between principal variables and non-principal variables can be exploited to speed up the search. For example, if a variable $x_2$ is highly correlated with another variable $x_1$, then we can generate values for $x_2$ based on the values of $x_1$. As shown in Fig 2, the curve fittings at time $t-1$ and $t$, denoted $CF_{t-1}$ and $CF_t$ respectively, are computed by a polynomial fitting strategy on the corresponding non-dominated set. Then, the relationship between variables in the new environment ($CF_{t+1}$) can be predicted using the last two consecutive $CF_{t-1}$ and $CF_t$.

$$move_t = CF_t + (CF_t - CF_{t-1}) \tag{6}$$

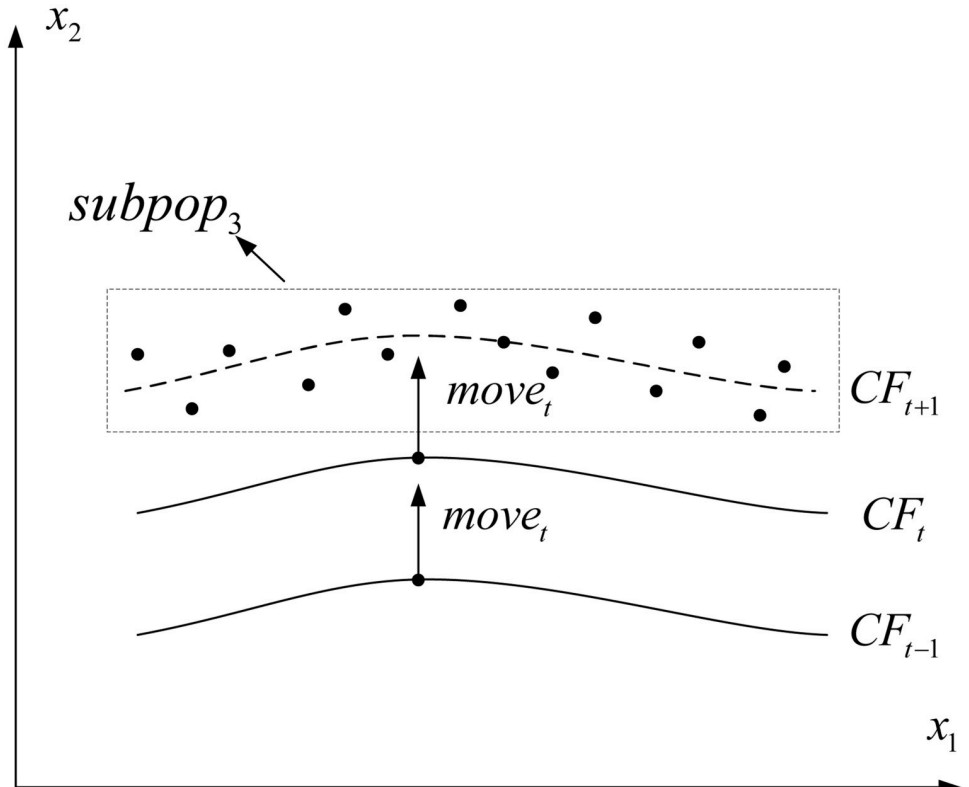

**Fig 2. Illustration of curve fitting-based strategy.**

Then, the possible curve fitting characteristic of at time $t + 1$ can be calculated as

$$CF_{t+1} = CF_t + move_t \tag{7}$$

In addition, individuals in the third subpopulation can be generated using the following formula,

$$Ind_{t+1} = CF_{t+1} + cr \times ND_p \tag{8}$$

where $cr$ is the compression radio, which ensures that the newly generated individual surrounds the curve fitting closely. $ND_p$ refers to the normal distribution based on the $pth$ variable, since this can make that the newly generated variables meet the characteristics of curve fitting as much as possible.

The implementation of this strategy is shown in Algorithm 4. Specifically, the the most principal variable is identified by the correlation matrix of variables, and the other variables are regards as non-principal variable. Then, for each non-principal variable, the corresponding values can be predicted by the curve fitting model which uses the values of the principal variable sampled from normal distribution. After, another subpopulation can be created by concatenating all the variables.

**Algorithm 4** Implementation details of Curve fitting-based strategy

```
1: Find the populations (Pop_t and Pop_{t-1}) at time t and t - 1,
respectively.
2: Compute the correlation matrix for each non-principal variable x_i
at time t - 1.
3: Estimate the new curve fitting feature for each x_i at time t + 1
according to Eq (7).
4: Create a subpopulation SubPop_2 by sampling from the decision space.
5: Calculate the bounds of x_i.
```

## 4 Experimental studies

This section evaluates the performance of the proposed algorithm through experimental studies. It includes details about benchmark functions, performance indicators, compared methods, parameter settings and numerical results.

### 4.1 Test instances

This work utilizes a set of recently proposed DF problems with various difficulties, such as variable linkage, disconnectivity, irregular POF shapes, and time-dependent geometries. All parameter settings keep the same with the suggestion according to the literature [58].

### 4.2 Performance indicators

This study employs three widely used performance indicators described as follows for evaluating the effectiveness of the proposed algorithm.

**4.2.1 Mean Inverted generational distance (MIGD).** The first performance indicator is MIGD, which is utilized to evaluate the convergence and diversity of solutions obtained by an algorithm, and the mathematical equation is provided as follows [56, 59].

$$IGD(POF_t^*, POF_t^{ob}) = \frac{\sum_{g \in POF_t^*} d(g, POF_t^{ob})}{|POF_t^*|} \tag{9}$$

where $POF_t^*$ is the true POF solutions, $POF_t^{ob}$ is a POF approximation, $d(g, POF_t^{ob})$ is the minimum Euclidian distance between $g$ and the points in $POF_t^{ob}$, and $|POF_t^*|$ is the number of

solution in $POF_t^*$. Then, the MIGD can be computed as

$$MIGD = \frac{\sum_{t \in T} IGD(POF_t^*, POF_t^{ob})}{|T|}, \tag{10}$$

where $T$ is a set of times instance and $|T|$ is the total number of changes in a run.

**4.2.2 Mean Schott's Spacing Metric (MSP).** The second performance indicator is the Schott's spacing metric, which is used to measure the distribution of the obtained solutions $POF_t^{ob}$ using the following formula:

$$SP(POF_t^{ob}) = \sqrt{\frac{1}{|POF_t^{ob}| - 1} \left( \sum_{i=1}^{|POF_t^{ob}|} (D_i - \bar{D}) \right)} \tag{11}$$

where $D_i$ represents the Euclidean distance between the $ith$ point in $POF_t^{ob}$ and its nearest point in $POF_t^{ob}$. $\bar{D}$ is the average value of $D_i$. The MSP can be defined as follows:

$$MSP = \frac{\sum_{t \in T} SP(POF_t^{ob})}{|T|}. \tag{12}$$

**4.2.3 Hypervolume metric.** The second performance indicator is Hypervolume (HV) [48, 53], which is a important metric for evaluating solutions. Different from the other indicators mentioned above, HV needs to set a reference vector dominated by any points in the $POF_t^*$.

$$HV_t = HV(POF_t^{ob}), \tag{13}$$

where $HV(POF_t^{ob})$ refers to the hypervolume [52] of set $POF_t^{ob}$. The reference point for the computation of hypervolume is $(z_j + 0.5, j = 1, \ldots, m)$, where $z_j$ is the maximum value of the $j$th objective of true POF. The MHV can be calculated as follows:

$$MHV = \frac{\sum_{t \in T} HV_t}{|T|}. \tag{14}$$

**4.2.4 T-test.** To determine whether the results obtained by the proposed algorithm are essentially difference from the results computed by other algorithms, the t-test at a 0.05 significance level is employed to check the experimental results of all optimization methods [60]. A $p$−value less than 0.05 indicates that the performance of two compared techniques is statistically different ($h = 1$), otherwise, there is no significant difference ($h = 0$). Meanwhile, the bottom of each Table summarizes the comparison results, ‡, † and ≀ indicate that the performance of FBP is better than, worse than and similar to that of the corresponding algorithm, respectively.

## 4.3 Compared algorithms

In this section, several existing approaches are selected to compare with the proposed technique. A brief description of these algorithms and parameter settings is summarized as follows.

**4.3.1 Population Prediction Strategy (PPS).** The main idea of PPS is to divide the PS/PF into two parts: population center and manifold. Autoregression (AR) model is adopted to predict the next population center based on a time series of historical population centers.

Similarly, historical manifolds are also used to predict new manifold. Then, A new population will be assembled based on the predicted population center and manifold [48].

**4.3.2 TrDMOEA.** TrDMOEA is an approach integrating transfer learning strategy and evolutionary algorithms to solve DMOPs. This main idea of this technique is that the agents at different times have different distributions for generating an effective search population. More details can be found in the literature [4].

**4.3.3 MOE.** MOE is a mixture-of-experts-based computation framework with multiple prediction mechanism for generating robust POS and enhancing the overall prediction quality in dealing with DMOPs. Experimental results illustrate that MOE has significant performance with respective to other dynamic optimization algorithms. More details can be found in the literature [61].

**4.3.4 MOEA/D-FD.** First-order difference model-based MOEA/D algorithm (MOEA/D-FD) [62] utilizes historical information to predict the location of the new POS after a change is detected. The new population is composed of two kinds of solutions: the old solutions and the predicted ones. The movement of population centroid defines a predicted direction. To make the new population diversified, evenly-distributed individuals selected from the previous population are used in the prediction.

**4.3.5 MOGOA.** The Grasshopper Optimisation Algorithm (GOA) models is proposed according to the behaviour of grasshopper swarms in nature, and a multi-objective version of Grasshopper optimization algorithm, MOGOA, is also designed for solving different multi-objective optimization problems. To enhance the distribution of solutions, an archive and a roulette wheel selection technique are integrated to the algorithm, and the individuals with uncrowded distance tend to be deleted for avoiding premature convergence. More details can be found in the literature [20].

**4.3.6 MOMVO.** The multi-verse optimization is proposed by imitating the white hole, black hole and wormhole mechanisms, which correspond to three different search strategies, exploration, exploitation, and local search, respectively. Meanwhile, a multi-objective version of multi-verse optimization, MOMVO, is also designed for solving different multi-objective optimization problems. In which, a leader selection strategy is utilized to choose the better agents from the archive, in addition, all the individuals will be ranked based on crowded distance with its neighbourhoods, and will be selected using the roulette wheel strategy for maintaining the convergence and diversity. More details can be found in the literature [21].

**4.3.7 MOALO.** The ALO algorithm, a new population-based optimization technique, is proposed by simulating the interaction and hunting behaviors of antlions in nature. Recently, it is also considered as an extended version, Multi-objective ant lion optimizer (MOALO), in which the non-dominated relationships and roulette wheel strategy are utilized to generating promising solutions. In addition, a set of benchmark functions and some constrained engineering design problems are cited to check the performance of MOALO. More details can be found in the literature [22].

**4.3.8 MSSA.** The SSA algorithm is designed based on the swarming behavior of salps when navigating and foraging in oceans for solving various optimization problems. Recently, it is also considered as an extended version, Multi-objective Salp Swarm Algorithm (MSSA), in which the guidance solution is selected from a set of non-dominated solutions based on ranking process and roulette wheel selection strategies, and the individuals with low rank tend to be deleted probability for maintaining the scale of archive. More details can be found in the literature [23].

## 4.4 Parameter settings

The parameters of the MOEAs considered in the experiment were referenced from their original papers. Some key parameters in these algorithms were set as follows.

**4.4.1 Population size.** The population size ($N$) in all the algorithms was set to 100. Around 1000 points were uniformly sampled from the true POF for computing the performance metrics in both bi- and three-objective cases.

**4.4.2 Other parameters.** All the parameters in the compared algorithms used the same settings as in their original studies.

Parameters in FBP: the degree of polynomial fit (*dpf*) is set to 2, the size of $SubPop_2$ was set to 0.4$N$, the parameters of the third strategy are seen [56].

**4.4.3 Stopping criterion and the number of executions.** Each algorithm terminates after a prespecified number of generations and should cover all possible changes. To minimize the effect of static optimization, we gave 50 generations for each algorithm before the first change occurs. The total number of generations was set to $3n_t \tau_t + 50$, which ensures there are $3n_t$ changes during the evolution. Additionally, each algorithm was executed 25 independent times on each test instance.

**4.4.4 Change detection.** For all the algorithms, a maximum number of 10% population are re-evaluated for change detection.

## 4.5 Experimental results

The severity of change ($n_t$) and the frequency of change ($\tau_t$) are two significant parameters in benchmark functions. To investigate the influence of these parameters on algorithms' performance, they are set to different values (5,10,20) in this section. Tables 1–9 summarize the numerical results obtained by different algorithms, and the best values are also highlighted in bold face.

The MIGD results of all the algorithms are recorded in Tables 1–3, and it can be seen that FBP has the best values compared with its peers for most of the benchmark functions. However, for two functions DF4 and DF8, FBP is not able to obtain the best value, but the difference is not large according to the statistical $p$-values. When the $\tau_t$ is set to 10, FBP generates the best result on DF1. Meanwhile, for different levels of $n_t$ and $\tau_t$, the proposed technique still can achieve the best result on majority of the functions. This shows that the designed prediction strategies can generate good population tracking the true POF closely in dynamic environments.

As shown in Tables 4–6, which summarizes the MHV values of all the algorithms, although FBP has great better MHV values than the other techniques on a majority of the problems, it is not effective enough for solving DF5, DF6 and DF11 based on the statistical $t-test$ results. In addition, MOE has little advantage over the others on DF9 and DF14. Therefore, the MHV metric further demonstrates that the proposed strategy responds to changes well.

It is observed from Tables 7–9, which lists MSP results obtained by all the algorithms, that although FBP can obtain the best solution on most of two bi-objective problems, e.g., DF1, DF5 and DF7, it seems ineffective in a few three objective problems, but the difference is not significant according to the statistical $p$-values. MOEA/D-FD obtains best distribution of solutions in other cases. MOEA/D-FD benefits from the even weights in its decomposition approach that improves the distribution of solutions. On the contrary, the other MOEAs utilizes dominance-based environmental selection approaches, which may not generate as uniform solutions as the decomposition-based technique, especially in three-objective problems. Besides that, well-distributed solutions does not mean that they approximate the true POF closely. MOEA/D-FD performs better than FBP in terms of MSP, but it is weaker than FBP on

**Table 1. Mean and standard deviation values of MIGD obtained by five algorithm for $(n_t, \tau_t) = (5,20)$.**

| Fun. | $(n_t, \tau_t)$ | MOEA/D-FD | TrDMOEA | MoE | PPS | FBP |
|------|------|-----------|---------|-----|-----|-----|
| DF1 | (5,20) | 1.179e-2(1.764e-4) | 1.777e-2(2.139e-3) | **6.941e-3(3.613e-4)** | 3.668e-1(7.186e-2) | 1.065e-2(7.992e-4) |
| | p | 1.761e-1 | 6.696e-11 | 3.339e-3 | 7.196e-11 | - |
| | h | 0 | 1 | 1 | 1 | - |
| DF2 | (5,20) | 1.073e-2(3.404e-4) | **6.565e-3(6.454e-4)** | 1.323e-2(4.886e-4) | 2.440e-1(5.131e-2) | 4.170e-2(3.022e-3) |
| | p | 3.324e-6 | 1.492e-6 | 6.356e-5 | 1.992e-7 | - |
| | h | 1 | 1 | 1 | 1 | - |
| DF3 | (5,20) | 4.606e-2(4.499e-3) | 5.734e-2(1.981e-2) | 1.364e-1(1.182e-4) | 1.797e-1(1.494e-1) | **1.882e-2(6.951e-3)** |
| | p | 7.088e-8 | 4.616e-10 | 1.067e-7 | 5.163e-9 | - |
| | h | 1 | 1 | 1 | 1 | - |
| DF4 | (5,20) | 1.186e-1(2.085e-3) | 5.872e-1(1.742e-3) | 1.066e+0(1.578e-3) | 1.370e-1(1.003e-2) | **9.724e-2(3.504e-3)** |
| | p | 1.373e-1 | 2.707e-1 | 5.494e-11 | 3.511e-1 | - |
| | h | 0 | 0 | 1 | 0 | - |
| DF5 | (5,20) | 2.027e-2(2.061e-4) | 2.808e-2(3.792e-4) | 1.533e+0(1.063e-3) | 3.723e-1(1.041e-1) | **1.541e-2(9.571e-4)** |
| | p | 2.151e-2 | 5.072e-10 | 6.735e-1 | 5.588e-10 | - |
| | h | 1 | 1 | 0 | 1 | - |
| DF6 | (5,20) | 4.514e+0(4.384e-1) | 9.798e-1(2.154e-1) | 2.473e+0(1.437e+0) | 6.897e+0(8.883e-1) | **4.675e-1(1.882e-2)** |
| | p | 1.260e-1 | 6.066e-11 | 2.838e-1 | 6.566e-11 | - |
| | h | 0 | 1 | 0 | 1 | - |
| DF7 | (5,20) | 8.858e-2(1.863e-2) | 3.829e-2(1.287e-3) | 5.978e+0(1.076e-2) | 6.720e-2(1.938e-2) | **1.066e-2(3.048e-4)** |
| | p | 8.993e-11 | 4.200e-10 | 3.020e-11 | 4.700e-9 | - |
| | h | 1 | 1 | 1 | 1 | - |
| DF8 | (5,20) | 5.631e-2(1.418e-3) | 8.208e-2(4.023e-4) | 1.546e-2(2.405e-4) | 4.636e-2(1.662e-3) | **3.715e-2(3.864e-4)** |
| | p | 1.597e-3 | 2.062e-1 | 2.170e-1 | 2.663e-1 | - |
| | h | 1 | 0 | 0 | 0 | - |
| DF9 | (5,20) | 8.535e-2(1.959e-2) | 9.792e-2(2.423e-3) | 1.109e+0(1.391e-3) | 5.431e-1(1.111e-1) | **9.557e-2(6.669e-3)** |
| | p | 8.500e-2 | 8.101e-10 | 4.062e-2 | 8.610e-10 | - |
| | h | 0 | 1 | 1 | 1 | - |
| DF10 | (5,20) | 1.877e-1(4.411e-2) | 2.804e-1(6.160e-3) | 1.541e-1(5.677e-3) | 1.931e-1(1.144e-2) | **1.078e-1(3.876e-3)** |
| | p | 2.236e-2 | 1.628e-2 | 2.154e-10 | 5.092e-8 | - |
| | h | 1 | 1 | 1 | 1 | - |
| DF11 | (5,20) | 6.514e-1(4.128e-4) | 2.846e-1(3.159e-2) | **9.541e-2(3.646e-4)** | 6.691e-1(2.447e-3) | 6.574e-1(1.225e-3) |
| | p | 2.905e-1 | 7.652e-5 | 6.528e-8 | 2.581e-1 | - |
| | h | 0 | 1 | 1 | 0 | - |
| DF12 | (5,20) | 8.731e-1(3.021e-2) | 3.266e-1(1.545e-2) | 2.904e-1(1.510e-3) | 3.123e-1(1.151e-2) | **2.794e-1(6.227e-3)** |
| | p | 5.533e-8 | 3.367e-5 | 5.494e-11 | 5.555e-2 | - |
| | h | 1 | 1 | 1 | 0 | - |
| DF13 | (5,20) | 2.530e-1(1.364e-2) | 1.659e-1(2.258e-3) | 2.522e+0(1.163e-2) | 4.148e-1(4.258e-2) | **1.608e-1(6.489e-3)** |
| | p | 9.833e-8 | 3.318e-1 | 9.919e-11 | 1.026e-10 | - |
| | h | 1 | 0 | 1 | 1 | - |
| DF14 | (5,20) | 1.282e-1(2.850e-3) | 7.204e-2(3.136e-4) | 1.039e+0(4.048e-3) | 1.552e-1(1.963e-2) | **5.720e-2(1.533e-3)** |
| | p | 3.255e-7 | 2.006e-4 | 1.287e-9 | 5.072e-10 | - |
| | h | 1 | 1 | 1 | 1 | - |
| | ‡/†/ℓ | 8/1/5 | 9/2/3 | 8/3/3 | 10/0/4 | - |

**Table 2. Mean and standard deviation values of MIGD obtained by five algorithm for $(n_t, \tau_t) = (10,10)$.**

| Fun. | $(n_t, \tau_t)$ | MOEA/D-FD | TrDMOEA | MoE | PPS | FBP |
|------|------|-----------|---------|-----|-----|-----|
| DF1 | (10,10) | 9.522e-3(1.371e-4) | 8.431e-2(9.136e-2) | 1.487e-2(1.322e-3) | 3.729e-1(6.416e-2) | **5.844e-3(2.682e-4)** |
| | $p$ | 1.606e-6 | 1.094e-10 | 3.338e-11 | 3.348e-11 | - |
| | $h$ | 1 | 1 | 1 | 1 | - |
| DF2 | (10,10) | 1.097e-2(2.093e-4) | **8.149e-3(5.478e-4)** | 3.718e-2(2.567e-3) | 2.261e-1(4.723e-2) | 3.937e-2(4.765e-3) |
| | $p$ | 2.126e-4 | 1.202e-8 | 3.953e-1 | 1.698e-8 | - |
| | $h$ | 1 | 1 | 0 | 1 | - |
| DF3 | (10,10) | 3.386e-2(2.023e-3) | 3.358e-2(1.542e-2) | 1.447e-1(9.352e-4) | 1.460e-1(1.323e-1) | **1.162e-2(5.596e-3)** |
| | $p$ | 1.311e-8 | 3.256e-7 | 5.573e-10 | 1.777e-10 | - |
| | $h$ | 1 | 1 | 1 | 1 | - |
| DF4 | (10,10) | 1.069e-1(1.686e-3) | 5.644e-1(1.407e-1) | 1.209e+0(3.958e-3) | 1.162e-1(1.222e-2) | **7.029e-2(2.230e-3)** |
| | $p$ | 1.221e-2 | 1.558e-8 | 3.020e-11 | 8.500e-1 | - |
| | $h$ | 1 | 1 | 1 | 0 | - |
| DF5 | (10,10) | 1.455e-2(2.501e-4) | 2.583e-2(4.359e-3) | 1.228e+0(2.129e-3) | 3.628e-1(9.444e-2) | **6.832e-3(7.967e-4)** |
| | $p$ | 4.686e-8 | 4.195e-10 | 3.324e-6 | 4.504e-11 | - |
| | $h$ | 1 | 1 | 1 | 1 | - |
| DF6 | (10,10) | 5.080e+0(5.327e-1) | 1.209e+0(2.704e-1) | 4.581e+0(6.182e-1) | 7.477e+0(7.384e-1) | **6.140e-1(4.412e-2)** |
| | $p$ | 4.675e-2 | 7.482e-2 | 1.087e-1 | 6.695e-11 | - |
| | $h$ | 1 | 0 | 0 | 1 | - |
| DF7 | (10,10) | 9.106e-2(1.418e-2) | 3.546e-2(9.378e-4) | 3.318e+0(1.465e-3) | 6.051e-2(1.616e-2) | **9.785e-3(4.370e-4)** |
| | $p$ | 3.020e-11 | 6.066e-11 | 3.020e-11 | 1.329e-10 | - |
| | $h$ | 1 | 1 | 1 | 1 | - |
| DF8 | (10,10) | 3.053e-2(2.051e-3) | 7.954e-2(7.347e-3) | 1.891e-2(6.910e-4) | 1.569e-2(1.487e-3) | **7.152e-3(5.012e-4)** |
| | $p$ | 4.504e-11 | 3.094e-6 | 6.528e-8 | 3.183e-3 | - |
| | $h$ | 1 | 1 | 1 | 1 | - |
| DF9 | (10,10) | 8.732e-2(1.473e-2) | 7.540e-2(1.771e-2) | 1.078e+0(3.807e-3) | 4.713e-1(1.280e-1) | **6.755e-2(5.406e-3)** |
| | $p$ | 5.298e-1 | 2.708e-2 | 3.770e-4 | 8.153e-11 | - |
| | $h$ | 0 | 1 | 1 | 1 | - |
| DF10 | (10,10) | 1.652e-1(3.667e-2) | 2.775e-1(1.289e-2) | 1.542e-1(4.819e-3) | 1.816e-1(1.075e-2) | **1.001e-1(3.921e-3)** |
| | $p$ | 1.628e-2 | 1.441e-2 | 7.958e-3 | 2.015e-8 | - |
| | $h$ | 1 | 1 | 1 | 1 | - |
| DF11 | (10,10) | 6.373e-1(3.384e-4) | 2.877e-1(1.657e-2) | **1.020e-1(1.237e-3)** | 6.551e-1(2.509e-3) | 6.436e-1(1.138e-3) |
| | $p$ | 5.395e-1 | 2.275e-5 | 8.101e-10 | 5.493e-1 | - |
| | $h$ | 0 | 1 | 1 | 0 | - |
| DF12 | (10,10) | 9.526e-1(1.738e-2) | 3.559e-1(4.370e-2) | 3.261e-1(4.656e-3) | 3.043e-1(9.512e-3) | **2.831e-1(1.074e-2)** |
| | $p$ | 3.831e-5 | 7.695e-8 | 2.380e-3 | 1.494e-1 | - |
| | $h$ | 1 | 1 | 1 | 0 | - |
| DF13 | (10,10) | 2.239e-1(5.951e-3) | 1.542e-1(7.866e-3) | 2.107e+0(1.301e-2) | 4.057e-1(2.940e-2) | **1.500e-1(4.638e-3)** |
| | $p$ | 3.770e-4 | 9.926e-2 | 3.020e-11 | 8.153e-11 | - |
| | $h$ | 1 | 0 | 1 | 1 | - |
| DF14 | (10,10) | 1.221e-1(4.208e-3) | 6.943e-2(3.387e-3) | 8.480e-1(3.783e-3) | 1.620e-1(2.177e-2) | **5.251e-2(1.619e-3)** |
| | $p$ | 7.727e-2 | 2.133e-5 | 4.118e-6 | 3.820e-10 | - |
| | $h$ | 0 | 1 | 1 | 1 | - |
| | ‡/†/≀ | 10/1/3 | 10/2/2 | 11/1/2 | 11/0/3 | - |

**Table 3. Mean and standard deviation values of MIGD obtained by five algorithm for $(n_t, \tau_t)$ = (10,20).**

| Fun. | $(n_t, \tau_t)$ | MOEA/D-FD | TrDMOEA | MoE | PPS | FBP |
|---|---|---|---|---|---|---|
| DF1 | (10,20) | 6.546e-3(1.322e-4) | 5.809e-2(1.091e-3) | **7.599e-3(1.148e-3)** | 2.479e-1(5.558e-2) | 6.461e-2(3.306e-4) |
| | p | 9.234e-1 | 5.011e-1 | 6.765e-5 | 1.254e-7 | - |
| | h | 0 | 0 | 1 | 1 | - |
| DF2 | (10,20) | 9.222e-3(1.670e-4) | **8.197e-3(2.152e-5)** | 1.060e-2(6.531e-4) | 1.508e-1(4.644e-2) | 3.182e-2(2.665e-3) |
| | p | 1.370e-3 | 7.599e-7 | 5.322e-3 | 1.492e-6 | - |
| | h | 1 | 1 | 1 | 1 | - |
| DF3 | (10,20) | 9.217e-2(1.145e-3) | 8.895e-2(1.175e-3) | 1.416e-1(3.469e-4) | 1.662e-1(1.457e-1) | **7.838e-2(9.461e-3)** |
| | p | 7.483e-2 | 5.395e-1 | 1.383e-2 | 5.607e-5 | - |
| | h | 0 | 0 | 1 | 1 | - |
| DF4 | (10,20) | 4.351e-1(5.528e-3) | 5.588e-1(1.801e-2) | 1.200e+0(9.368e-4) | 4.808e-1(1.193e-2) | **4.328e-1(3.871e-3)** |
| | p | 9.234e-1 | 7.978e-2 | 6.283e-6 | 4.825e-1 | - |
| | h | 0 | 0 | 1 | 0 | - |
| DF5 | (10,20) | 2.580e-2(1.771e-4) | 3.118e-2(2.894e-4) | 1.221e+0(6.439e-4) | 1.044e-1(2.204e-2) | **2.219e-2(4.738e-4)** |
| | p | 6.377e-1 | 1.508e-1 | 4.060e-2 | 2.669e-5 | - |
| | h | 1 | 0 | 1 | 1 | - |
| DF6 | (10,20) | 3.235e+0(1.072e+0) | 1.953e+0(2.570e-1) | 3.771e+0(6.838e-1) | 4.032e+0(6.981e-1) | **3.822e-1(3.328e-2)** |
| | p | 5.943e-1 | 2.839e-4 | 1.958e-1 | 8.891e-10 | - |
| | h | 0 | 1 | 0 | 1 | - |
| DF7 | (10,20) | 1.241e-1(1.096e-2) | 7.679e-2(9.822e-3) | 3.310e+0(1.900e-4) | 8.795e-2(8.222e-3) | **5.523e-2(3.420e-4)** |
| | p | 1.407e-4 | 2.416e-2 | 3.020e-11 | 5.084e-3 | - |
| | h | 1 | 0 | 1 | 1 | - |
| DF8 | (10,20) | 1.374e-1(3.063e-3) | 8.472e-2(2.789e-5) | **1.590e-2(6.270e-4)** | 1.390e-1(3.522e-3) | 1.302e-1(1.222e-3) |
| | p | 6.843e-1 | 1.907e-1 | 6.356e-5 | 7.958e-1 | - |
| | h | 0 | 0 | 1 | 0 | - |
| DF9 | (10,20) | 7.396e-2(1.394e-3) | 6.534e-2(1.396e-2) | 1.057e+0(5.904e-4) | 2.970e-1(1.167e-1) | **5.778e-2(3.320e-3)** |
| | p | 7.172e-1 | 9.047e-2 | 1.031e-2 | 1.464e-10 | - |
| | h | 0 | 0 | 1 | 1 | - |
| DF10 | (10,20) | 3.147e-1(1.227e-2) | 2.867e-1(1.412e-2) | 2.442e-1(3.273e-3) | 2.491e-1(7.535e-3) | **2.068e-1(4.280e-3)** |
| | p | 1.335e-1 | 9.470e-1 | 9.000e-1 | 9.705e-1 | - |
| | h | 0 | 0 | 0 | 0 | - |
| DF11 | (10,20) | 7.480e-1(3.787e-4) | 2.507e-1(2.641e-2) | **9.407e-2(2.561e-4)** | 7.639e-1(1.988e-3) | 7.561e-1(1.703e-3) |
| | p | 5.692e-1 | 1.148e-7 | 3.020e-11 | 5.793e-1 | - |
| | h | 0 | 1 | 1 | 0 | - |
| DF12 | (10,20) | 9.348e-1(2.987e-2) | 3.369e-1(1.431e-2) | **3.039e-1(3.915e-3)** | 3.099e-1(1.016e-2) | 3.128e-1(6.493e-3) |
| | p | 6.121e-10 | 6.377e-3 | 2.002e-6 | 3.555e-1 | - |
| | h | 1 | 1 | 1 | 0 | - |
| DF13 | (10,20) | 2.782e-1(1.351e-2) | 1.715e-1(2.208e-3) | 2.049e+0(9.606e-3) | 2.996e-1(1.908e-2) | **1.705e-1(5.669e-3)** |
| | p | 1.429e-8 | 3.871e-1 | 2.610e-10 | 2.592e-7 | - |
| | h | 1 | 0 | 1 | 1 | - |
| DF14 | (10,20) | 1.427e-1(4.208e-3) | 7.995e-2(7.074e-3) | 8.340e-1(2.587e-3) | 1.095e-1(9.542e-3) | **6.793e-2(2.561e-3)** |
| | p | 1.429e-8 | 4.427e-3 | 7.394e-1 | 1.0278e-6 | - |
| | h | 1 | 1 | 0 | 1 | - |
| | ‡/†/≀ | 5/1/8 | 3/2/9 | 6/5/3 | 9/0/5 | - |

the other two indicators, i.e., MIGD and MHV, which are more reliable to distinguish between algorithms in terms of the overall performance.

As described before, it is obvious that the frequency of changes exerts a certain influence on algorithms' performance. In three-objective functions, frequent changes increase the difficulty

**Table 4. Mean and standard deviation values of MHV obtained by five algorithms for $(n_t, \tau_t) = (5,20)$.**

| Fun. | $(n_t, \tau_t)$ | MOEA/D-FD | TrDMOEA | MoE | PPS | FBP |
|---|---|---|---|---|---|---|
| DF1 | (5,20) | 1.659e+0(7.233e-4) | 1.637e+0(2.276e-3) | 1.836e-2(9.513e-4) | 9.759e-1(7.666e-2) | **1.660e+0(4.520e-3)** |
| | p | 9.117e-1 | 1.055e-1 | 6.414e-7 | 3.497e-9 | - |
| | h | 0 | 0 | 1 | 1 | - |
| DF2 | (5,20) | 1.892e+0(8.087e-4) | **1.904e+0(8.154e-5)** | 3.805e-2(8.411e-4) | 1.390e+0(8.652e-2) | 1.796e+0(8.445e-3) |
| | p | 4.444e-7 | 4.200e-10 | 7.216e-1 | 3.324e-6 | - |
| | h | 1 | 1 | 0 | 1 | - |
| DF3 | (5,20) | 1.522e+0(2.334e-2) | 1.537e+0(5.905e-3) | 4.779e-1(3.447e-3) | 1.256e+0(2.644e-1) | **1.604e+0(1.667e-2)** |
| | p | 1.273e-2 | 6.842e-1 | 3.254e-3 | 1.091e-5 | - |
| | h | 1 | 0 | 1 | 1 | - |
| DF4 | (5,20) | 7.408e+0(7.686e-3) | 7.183e+0(5.905e-3) | 7.336e-1(7.610e-3) | 7.177e+0(6.985e-2) | **7.594e+0(1.718e-2)** |
| | p | 6.952e-1 | 5.298e-1 | 4.872e-5 | 5.692e-1 | - |
| | h | 0 | 0 | 1 | 0 | - |
| DF5 | (5,20) | 1.716e+0(5.630e-4) | 1.741e+0(4.824e-3) | **9.897e+0(8.611e-3)** | 1.055e+0(1.321e-1) | 1.728e+0(2.713e-3) |
| | p | 5.971e-5 | 6.358e-7 | 6.824e-8 | 2.154e-10 | - |
| | h | 1 | 1 | 1 | 1 | - |
| DF6 | (5,20) | 1.296e+0(5.173e-2) | 9.580e-1(2.451e-1) | **2.173e+2(1.918e+2)** | 3.437e-2(3.222e-2) | 1.241e+0(1.373e-2) |
| | p | 9.705e-1 | 1.496e-1 | 8.687e-1 | 9.050e-8 | - |
| | h | 0 | 0 | 0 | 1 | - |
| DF7 | (5,20) | 3.376e+0(1.475e-2) | 3.412e+0(9.481e-3) | 1.911e+0(9.785e-1) | 3.271e+0(5.757e-2) | **3.466e+0(2.495e-3)** |
| | p | 2.052e-3 | 2.170e-1 | 7.454e-6 | 2.266e-3 | - |
| | h | 1 | 0 | 1 | 1 | - |
| DF8 | (5,20) | 1.776e+0(7.908e-4) | 1.728e+0(5.059e-4) | 1.709e-1(7.846e-4) | 1.761e+0(4.778e-3) | **1.788e+0(8.741e-4)** |
| | p | 3.644e-2 | 7.645e-4 | 3.179e-10 | 5.368e-2 | |
| | h | 1 | 1 | 1 | 0 | - |
| DF9 | (5,20) | 1.555e+0(2.818e-2) | 1.626e+0(7.786e-3) | **9.062e+0(1.181e-2)** | 8.214e-1(1.029e-1) | 1.517e+0(1.751e-2) |
| | p | 8.500e-1 | 3.158e-4 | 7.456e-1 | 1.777e-10 | - |
| | h | 0 | 1 | 0 | 1 | - |
| DF10 | (5,20) | 1.357e+0(9.566e-3) | 1.370e+0(2.002e-2) | 5.961e-1(1.547e-1) | 1.067e+0(3.106e-2) | **1.377e+0(9.617e-3)** |
| | p | 5.201e-1 | 7.287e-3 | 8.246e-10 | 2.602e-8 | - |
| | h | 0 | 1 | 1 | 1 | - |
| DF11 | (5,20) | 2.960e-1(1.461e-3) | 8.041e-1(5.668e-2) | **4.182e+0(2.323e-2)** | 3.386e-1(4.894e-3) | 3.570e-1(2.654e-3) |
| | p | 1.488e-1 | 1.247e-4 | 2.604e-1 | 6.100e-1 | - |
| | h | 0 | 1 | 0 | 0 | - |
| DF12 | (5,20) | 3.262e+0(3.577e-2) | **3.561e+0(1.336e-3)** | 2.403e+0(1.292e-1) | 3.084e+0(3.766e-2) | 3.368e+0(1.022e-2) |
| | p | 9.117e-1 | 1.096e-5 | 4.674e-6 | 1.041e-4 | - |
| | h | 0 | 1 | 1 | 1 | - |
| DF13 | (5,20) | 6.736e+0(1.178e-2) | 7.080e+0(3.604e-2) | 3.744e+2(1.391e+2) | 5.536e+0(2.734e-1) | **7.155e+0(3.007e-2)** |
| | p | 1.370e-3 | 1.907e-1 | 5.476e-10 | 9.063e-8 | - |
| | h | 1 | 0 | 1 | 1 | - |
| DF14 | (5,20) | 9.132e-1(4.500e-3) | 1.115e+0(1.136e-2) | **3.049e+1(8.788e-1)** | 8.826e-1(3.799e-2) | 1.072e+0(2.612e-3) |
| | p | 1.087e-1 | 7.506e-1 | 2.375e-2 | 3.032e-2 | - |
| | h | 0 | 0 | 1 | 1 | - |
| | ‡/†/$\ell$ | 5/1/8 | 3/4/7 | 8/2/4 | 11/0/3 | - |

**Table 5. Mean and standard deviation values of MHV obtained by five algorithms for $(n_t, \tau_t) = (10,10)$.**

| Fun. | $(n_t, \tau_t)$ | MOEA/D-FD | TrDMOEA | MoE | PPS | FBP |
|------|------|-----------|---------|-----|-----|-----|
| DF1 | (10,10) | 1.661e+0(3.713e-4) | 1.664e+0(1.427e-1) | 4.188e-2(2.920e-3) | 9.604e-1(7.314e-2) | **1.668e+0(9.101e-4)** |
| | $p$ | 1.857e-9 | 4.075e-11 | 3.432e-10 | 1.174e-9 | - |
| | $h$ | 1 | 1 | 1 | 1 | - |
| DF2 | (10,10) | 1.892e+0(5.494e-4) | **1.900e+0(1.105e-3)** | 1.116e-1(9.048e-3) | 1.423e+0(8.355e-2) | 1.798e+0(1.307e-2) |
| | $p$ | 2.610e-10 | 6.066e-11 | 4.210e-11 | 2.572e-7 | - |
| | $h$ | 1 | 1 | 1 | 1 | - |
| DF3 | (10,10) | 1.559e+0(1.185e-2) | 1.611e+0(4.851e-2) | 4.200e-1(6.448e-3) | 1.321e+0(2.166e-1) | **1.614e+0(1.295e-2)** |
| | $p$ | 8.663e-5 | 3.368e-5 | 6.211e-9 | 1.430e-5 | - |
| | $h$ | 1 | 1 | 1 | 1 | - |
| DF4 | (10,10) | 7.400e+0(3.674e-3) | 7.528e+0(5.462e-1) | 7.516e-1(1.778e-2) | 7.230e+0(7.289e-2) | **7.588e+0(1.027e-2)** |
| | $p$ | 9.470e-1 | 1.858e-1 | 5.414e-8 | 5.106e-1 | - |
| | $h$ | 0 | 0 | 1 | 0 | - |
| DF5 | (10,10) | 1.720e+0(4.886e-4) | 1.753e+0(1.809e-2) | **7.931e+0(9.442e-3)** | 1.067e+0(1.035e-1) | 1.734e+0(1.575e-3) |
| | $p$ | 5.072e-10 | 6.466e-11 | 4.504e-10 | 6.696e-11 | - |
| | $h$ | 1 | 1 | 1 | 1 | - |
| DF6 | (10,10) | 1.231e+0(8.906e-3) | 1.115e+0(7.465e-2) | **3.577e+2(6.934e+1)** | 2.707e-2(2.205e-2) | 1.214e-1(3.713e-2) |
| | $p$ | 1.511e-7 | 4.969e-9 | 6.751e-9 | 1.931e-7 | - |
| | $h$ | 1 | 1 | 1 | 1 | - |
| DF7 | (10,10) | 3.374e+0(1.306e-2) | 3.426e+0(1.283e-2) | 1.805e+2(2.524e+0) | 3.283e+0(5.049e-2) | **3.465e+0(2.298e-3)** |
| | $p$ | 6.952e-1 | 1.120e-1 | 5.799e-5 | 8.315e-3 | - |
| | $h$ | 0 | 0 | 1 | 1 | - |
| DF8 | (10,10) | 1.774e-2(5.440e-4) | 1.735e+0(1.954e-2) | 1.739e-1(2.639e-3) | 1.750e+0(6.110e-3) | **1.784e+0(1.653e-3)** |
| | $p$ | 9.117e-1 | 3.207e-7 | 8.314e-1 | 2.707e-1 | |
| | $h$ | 0 | 1 | 0 | 0 | - |
| DF9 | (10,10) | 1.548e+0(2.561e-2) | 1.645e+0(4.368e-2) | **8.884e+0(4.240e-2)** | 8.946e-1(1.307e-1) | 1.571e+0(1.172e-2) |
| | $p$ | 1.734e-9 | 8.516e-11 | 1.354e-9 | 4.504e-11 | - |
| | $h$ | 1 | 1 | 1 | 1 | - |
| DF10 | (10,10) | 1.358e+0(1.102e-2) | **1.908e+0(1.365e-3)** | 1.140e+0(3.321e-1) | 1.087e+0(2.755e-2) | 1.379e+0(1.174e-2) |
| | $p$ | 4.733e-1 | 2.754e-3 | 6.414e-1 | 1.174e-9 | - |
| | $h$ | 0 | 1 | 0 | 1 | - |
| DF11 | (10,10) | 3.097e-1(1.089e-3) | 7.456e-1(4.820e-3) | **4.339e+0(6.078e-2)** | 3.514e-1(5.745e-3) | 3.714e-1(2.397e-3) |
| | $p$ | 2.052e-1 | 4.978e-4 | 4.604e-5 | 5.493e-1 | - |
| | $h$ | 0 | 1 | 1 | 0 | - |
| DF12 | (10,10) | 3.414e-1(1.211e-2) | **3.560e+0(1.743e-5)** | 2.306e+0(5.445e-1) | 3.200e+0(4.098e-2) | 3.458e+0(1.013e-2) |
| | $p$ | 7.394e-1 | 1.418e-5 | 8.726e-7 | 2.154e-6 | - |
| | $h$ | 0 | 1 | 1 | 1 | - |
| DF13 | (10,10) | 6.674e+0(9.605e-3) | 7.044e+0(3.101e-2) | 3.067e+0(3.395e-2) | 5.573e+0(2.023e-1) | **7.105e+0(2.803e-2)** |
| | $p$ | 1.953e-3 | 1.004e-3 | 5.498e-8 | 5.186e-7 | - |
| | $h$ | 1 | 1 | 1 | 1 | - |
| DF14 | (10,10) | 9.175e-1(7.023e-3) | 1.116e+0(1.702e-2) | **2.379e+1(8.617e-1)** | 8.552e-1(4.905e-2) | 1.073e+0(1.995e-3) |
| | $p$ | 9.334e-2 | 7.172e-1 | 4.350e-3 | 1.564e-2 | - |
| | $h$ | 0 | 0 | 1 | 1 | - |
| | ‡/†/ℓ | 5/2/7 | 4/7/3 | 7/5/2 | 11/0/3 | - |

**Table 6. Mean and standard deviation values of MHV obtained by five algorithms for $(n_t, \tau_t) = (10,20)$.**

| Fun. | $(n_t, \tau_t)$ | MOEA/D-FD | TrDMOEA | MoE | PPS | FBP |
|------|------|-----------|---------|-----|-----|-----|
| DF1 | (10,20) | 1.585e+0(5.552e-4) | 1.534e+0(2.841e-3) | 1.808e-2(2.391e-3) | 1.498e+0(1.010e-1) | **1.592e+0(1.755e-3)** |
| | $p$ | 6.972e-3 | 1.252e-7 | 8.214e-8 | 3.646e-8 | - |
| | $h$ | 1 | 1 | 1 | 1 | - |
| DF2 | (10,20) | 1.896e+0(3.873e-4) | **1.901e+0(4.106e-4)** | 2.995e-2(1.905e-3) | 1.582e+0(8.767e-2) | 1.822e+0(7.472e-3) |
| | $p$ | 7.697e-4 | 4.686e-8 | 8.562e-7 | 8.292e-6 | - |
| | $h$ | 1 | 1 | 1 | 1 | - |
| DF3 | (10,20) | 1.461e+0(1.215e-2) | 1.520e+0(1.832e-2) | 4.048e-1(3.253e-3) | 1.312e+0(2.375e-1) | **1.521e+0(2.277e-2)** |
| | $p$ | 7.088e-8 | 1.013e-6 | 7.257e-11 | 6.121e-10 | - |
| | $h$ | 1 | 1 | 1 | 1 | - |
| DF4 | (10,20) | 7.850e+0(4.671e-3) | 7.782e+0(2.463e-2) | 6.952e-1(4.648e-3) | 7.671e+0(6.295e-2) | **7.984e+0(1.579e-2)** |
| | $p$ | 8.073e-1 | 8.534e-1 | 4.467e-7 | 6.375e-1 | - |
| | $h$ | 0 | 0 | 1 | 0 | - |
| DF5 | (10,20) | 1.718e+0(4.654e-4) | 1.741e+0(2.519e-4) | **7.890e+0(4.997e-3)** | 1.536e+0(4.837e-2) | 1.732e+0(1.038e-3) |
| | $p$ | 6.356e-5 | 3.835e-6 | 5.671e-5 | 2.034e-9 | - |
| | $h$ | 1 | 1 | 1 | 1 | - |
| DF6 | (10,20) | 1.693e+0(6.972e-3) | 9.867e-1(7.302e-2) | **2.827e+2(8.503e+1)** | 1.983e-1(1.478e-1) | 1.699e+0(1.579e-2) |
| | $p$ | 2.457e-1 | 3.279e-6 | 2.414e-6 | 1.654e-8 | - |
| | $h$ | 0 | 1 | 1 | 1 | - |
| DF7 | (10,20) | 3.295e+0(1.129e-2) | 3.285e+0(3.334e-2) | 1.743e+1(1.940e+0) | 3.219e+0(3.223e-2) | **3.378e+0(3.420e-4)** |
| | $p$ | 5.084e-3 | 1.171e-2 | 9.341e-4 | 6.549e-4 | - |
| | $h$ | 1 | 1 | 1 | 1 | - |
| DF8 | (10,20) | 1.775e+0(7.223e-4) | 1.724e+0(3.578e-3) | 1.715e-1(7.671e-4) | 1.760e+0(5.633e-3) | **1.786e+0(1.258e-3)** |
| | $p$ | 2.170e-1 | 2.530e-4 | 7.338e-4 | 3.112e-1 | - |
| | $h$ | 0 | 1 | 1 | 0 | - |
| DF9 | (10,20) | 1.593e+0(2.881e-2) | 1.632e+0(3.032e-2) | **8.619e+0(2.569e-2)** | 1.161e+0(1.610e-1) | 1.606e+0(9.745e-3) |
| | $p$ | 4.464e-1 | 7.562e-3 | 6.751e-9 | 2.372e-10 | - |
| | $h$ | 0 | 1 | 1 | 1 | - |
| DF10 | (10,20) | 1.676e+0(9.811e-3) | 1.595e+0(4.367e-2) | 6.541e-1(8.160e-2) | 1.606e+0(2.226e-2) | **1.759e+0(6.948e-3)** |
| | $p$ | 5.493e-1 | 3.112e-1 | 5.214e-3 | 3.183e-1 | - |
| | $h$ | 0 | 0 | 1 | 0 | - |
| DF11 | (10,20) | 1.730e-1(6.925e-4) | 8.559e-2(1.093e-2) | **4.157e+0(2.013e-2)** | 2.013e-1(4.002e-3) | 2.094e-1(3.150e-3) |
| | $p$ | 2.051e-3 | 3.780e-7 | 4.211e-7 | 5.011e-1 | - |
| | $h$ | 1 | 0 | 1 | 0 | - |
| DF12 | (10,20) | 3.454e+0(3.233e-2) | **3.558e+0(1.486e-3)** | 1.826e+0(1.742e-1) | 3.208e+0(3.708e-2) | 3.470e+0(5.415e-3) |
| | $p$ | 9.626e-2 | 4.857e-5 | 3.517e-5 | 1.249e-5 | - |
| | $h$ | 0 | 1 | 1 | 1 | - |
| DF13 | (10,20) | 6.847e+0(1.276e-2) | 7.208e+0(5.115e-2) | 2.924e+2(1.759e+2) | 6.398e+0(1.300e-1) | **7.277e+0(2.841e-2)** |
| | $p$ | 7.200e-5 | 5.298e-1 | 5.917e-5 | 3.571e-6 | - |
| | $h$ | 1 | 0 | 1 | 1 | - |
| DF14 | (10,20) | 9.242e-1(8.952e-3) | 1.098e+0(7.441e-3) | **2.210e+1(7.654e-1)** | 1.015e+0(1.820e-2) | 1.100e+0(2.159e-3) |
| | $p$ | 1.501e-2 | 9.352e-1 | 2.341e-3 | 2.398e-1 | - |
| | $h$ | 1 | 0 | 1 | 0 | - |
| | ‡/†/ℓ | 7/1/6 | 5/4/5 | 9/5/0 | 9/0/5 | - |

**Table 7. Mean and standard deviation values of MSP obtained by five algorithms for $(n_t, \tau_t) = (5,20)$.**

| Fun. | $(n_t, \tau_t)$ | MOEA/D-FD | TrDMOEA | MoE | PPS | FBP |
|------|------|------|------|------|------|------|
| DF1 | (5,20) | 9.007e-3(1.134e-4) | 1.482e-2(7.141e-3) | **4.320e-3(9.857e-5)** | 1.002e-1(3.494e-2) | 1.141e-2(3.280e-3) |
| | p | 3.183e-1 | 7.616e-3 | 5.714e-1 | 1.094e-10 | - |
| | h | 0 | 1 | 0 | 1 | - |
| DF2 | (5,20) | 1.142e-2(2.119e-4) | 8.111e-2(5.302e-2) | **8.417e-3(1.976e-4)** | 7.540e-2(3.698e-2) | 1.440e-1(1.658e-2) |
| | p | 3.020e-11 | 1.373e-1 | 3.674e-5 | 5.188e-2 | - |
| | h | 1 | 0 | 1 | 0 | - |
| DF3 | (5,20) | 1.180e-2(6.091e-4) | 1.914e-1(5.276e-2) | 4.079e-2(2.568e-4) | 6.683e-1(4.728e-1) | **2.982e-2(1.264e-2)** |
| | p | 8.684e-3 | 8.838e-7 | 6.414e-1 | 1.094e-10 | - |
| | h | 1 | 1 | 0 | 1 | - |
| DF4 | (5,20) | 8.899e-2(8.435e-3) | 1.926e-1(1.559e-3) | **1.538e-2(3.186e-4)** | 2.953e+0(8.708e+0) | 1.441e-1(6.296e-2) |
| | p | 8.120e-4 | 8.469e-4 | 4.642e-4 | 8.684e-3 | - |
| | h | 1 | 1 | 1 | 1 | - |
| DF5 | (5,20) | **1.036e-2(1.994e-4)** | 2.973e-2(1.921e-2) | 2.171e-2(6.409e-3) | 3.986e-1(2.958e-1) | 2.213e-2(6.724e-3) |
| | p | 1.031e-2 | 6.401e-1 | 7.380e-1 | 4.182e-9 | - |
| | h | 1 | 0 | 0 | 1 | - |
| DF6 | (5,20) | **2.009e-1(5.631e-2)** | 3.859e+0(1.792e+0) | 6.104e-1(1.298e-1) | 6.470e+1(4.073e+1) | 2.831e+0(4.179e-1) |
| | p | 3.474e-10 | 1.054e-3 | 4.579e-4 | 9.533e-7 | - |
| | h | 1 | 1 | 1 | 1 | - |
| DF7 | (5,20) | 2.961e-2(1.732e-3) | 1.372e+3(5.626e+1) | 1.488e+1(1.146e+0) | 2.068e-2(3.954e-3) | **7.480e-3(2.149e-4)** |
| | p | 3.020e-11 | 4.504e-11 | 5.641e-10 | 2.154e-10 | - |
| | h | 1 | 1 | 1 | 1 | - |
| DF8 | (5,20) | 1.730e-2(7.035e-4) | 6.123e-2(4.043e-2) | **1.630e-2(1.159e-3)** | 7.443e-2(2.676e-2) | 2.027e-2(5.085e-3) |
| | p | 1.537e-1 | 6.412e-1 | 3.587e-1 | 1.407e-4 | - |
| | h | 0 | 0 | 0 | 1 | - |
| DF9 | (5,20) | **1.156e-2(1.397e-3)** | 2.522e-1(8.904e-2) | 3.625e-2(2.658e-3) | 8.843e-1(4.800e-1) | 2.286e-2(4.963e-2) |
| | p | 3.020e-11 | 2.770e-1 | 8.502e-1 | 3.831e-5 | - |
| | h | 1 | 0 | 0 | 1 | - |
| DF10 | (5,20) | **3.024e-2(1.648e-2)** | 3.644e-1(1.845e-1) | 2.684e-2(1.431e-3) | 7.955e-1(2.165e-1) | 9.794e-1(1.545e-3) |
| | p | 3.020e-11 | 9.593e-11 | 1.249e-6 | 1.003e-3 | - |
| | h | 1 | 1 | 1 | 1 | - |
| DF11 | (5,20) | **2.413e-2(4.173e-4)** | 7.066e-2(1.358e-2) | 6.072e-2(3.346e-4) | 5.470e-2(4.152e-3) | 8.273e-2(1.377e-2) |
| | p | 3.020e-11 | 4.503e-7 | 2.178–9 | 5.678e-1 | - |
| | h | 1 | 1 | 1 | 0 | - |
| DF12 | (5,20) | **1.874e-2(8.696e-3)** | 3.253e-1(9.878e-3) | 5.243e-2(1.158e-3) | 7.505e-1(1.077e-1) | 6.305e-1(2.762e-2) |
| | p | 3.020e-11 | 1.070e-9 | 8.245e-8 | 8.500e-2 | - |
| | h | 1 | 1 | 1 | 0 | - |
| DF13 | (5,20) | **1.562e-1(4.104e-3)** | 3.645e-1(1.788e-1) | 4.294e-1(4.751e-2) | 2.587e+0(7.769e-1) | 9.457e-1(3.241e-2) |
| | p | 6.912e-4 | 4.553e-1 | 6.241e-3 | 1.167e-5 | - |
| | h | 1 | 0 | 1 | 1 | - |
| DF14 | (5,20) | **1.658e-2(5.116e-4)** | 3.095e-1(5.030e-2) | 1.201e-1(1.787e-3) | 3.700e-1(1.458e-1) | 6.810e-1(1.404e-2) |
| | p | 3.020e-11 | 6.787e-2 | 9.271e-10 | 1.297e-1 | - |
| | h | 1 | 0 | 1 | 0 | - |
| | ‡/†/ℓ | 4/7/2 | 5/3/6 | 1/8/5 | 9/1/4 | - |

**Table 8. Mean and standard deviation values of MSP obtained by five algorithms for $(n_t, \tau_t) = (10,10)$.**

| Fun. | $(n_t, \tau_t)$ | MOEA/D-FD | TrDMOEA | MoE | PPS | FBP |
|---|---|---|---|---|---|---|
| DF1 | (10,10) | 8.153e-3(1.007e-4) | 1.155e-2(6.465e-3) | 6.736e-3(2.778e-4) | 1.110e-1(3.643e-2) | **5.810e-3(1.984e-3)** |
| | $p$ | 3.020e-11 | 1.410e-9 | 7.515e-10 | 7.389e-11 | - |
| | $h$ | 1 | 1 | 1 | 1 | - |
| DF2 | (10,10) | **1.157e-2(2.434e-4)** | 9.109e-2(1.584e-2) | 1.407e-2(1.274e-3) | 7.791e-2(2.159e-2) | 1.309e-1(2.166e-2) |
| | $p$ | 3.020e-11 | 1.453e-1 | 2.017e-1 | 1.120e-1 | - |
| | $h$ | 1 | 0 | 0 | 0 | - |
| DF3 | (10,10) | 1.488e-2(4.903e-4) | 1.934e-1(1.176e-1) | 3.426e-2(4.130e-4) | 5.324e-1(3.303e-1) | **1.426e-2(7.014e-3)** |
| | $p$ | 3.020e-11 | 1.302e-3 | 6.717e-8 | 8.153e-11 | - |
| | $h$ | 1 | 1 | 1 | 1 | - |
| DF4 | (10,10) | 9.628e-2(9.741e-3) | 2.801e-1(2.005e-2) | **1.707e-2(7.243e-4)** | 6.327e-1(2.152e+0) | 7.809e-2(3.562e-2) |
| | $p$ | 7.245e-2 | 2.283e-2 | 5.207e-3 | 1.681e-4 | - |
| | $h$ | 0 | 1 | 1 | 1 | - |
| DF5 | (10,10) | 1.015e-2(1.539e-4) | 2.073e-1(1.432e-1) | 2.565e-2(7.398e-3) | 3.102e-1(1.535e-1) | **9.507e-3(3.978e-3)** |
| | $p$ | 3.020e-11 | 3.097e-1 | 3.418e-10 | 1.329e-10 | - |
| | $h$ | 1 | 0 | 1 | 1 | - |
| DF6 | (10,10) | **1.569e-1(2.206e-2)** | 3.678e+0(3.714e-1) | 6.180e-1(1.584e-1) | 7.450e+1(3.365e+1) | 9.134e+0(4.567e-1) |
| | $p$ | 3.690e-11 | 3.010e-7 | 8.715e-9 | 5.106e-1 | - |
| | $h$ | 1 | 1 | 1 | 0 | - |
| DF7 | (10,10) | 2.810e-2(7.358e-4) | 1.906e+2(1.134e+1) | 1.351e+1(2.711e+0) | 1.812e-2(2.106e-3) | **7.677e-3(2.650e-4)** |
| | $p$ | 3.770e-4 | 4.616e-10 | 6.185e-7 | 6.121e-10 | - |
| | $h$ | 1 | 1 | 1 | 1 | - |
| DF8 | (10,10) | 1.951e-2(8.917e-3) | 1.360e-1(1.541e-1) | 2.326e-2(4.328e-3) | 9.125e-2(3.940e-2) | **1.933e-2(4.811e-3)** |
| | $p$ | 5.462e-6 | 1.759e-1 | 5.971e-5 | 2.597e-5 | - |
| | $h$ | 1 | 0 | 1 | 1 | - |
| DF9 | (10,10) | **1.168e-2(1.170e-3)** | 4.862e-1(1.173e-1) | 5.036e-2(8.679e-3) | 6.686e-1(3.574e-1) | 1.893e-1(3.860e-2) |
| | $p$ | 3.020e-11 | 3.790e-1 | 4.065e-10 | 1.606e-6 | - |
| | $h$ | 1 | 0 | 1 | 1 | - |
| DF10 | (10,10) | **3.947e-2(2.235e-2)** | 3.057e-1(1.941e-2) | 3.375e-2(4.431e-3) | 9.253e-1(2.448e-1) | 9.422e-1(1.463e-2) |
| | $p$ | 3.020e-11 | 4.349e-11 | 2.107e-9 | 2.608e-2 | - |
| | $h$ | 1 | 1 | 1 | 1 | - |
| DF11 | (10,10) | **2.395e-2(3.900e-4)** | 1.288e-1(1.769e-2) | 6.122e-2(8.664e-4) | 5.418e-2(3.689e-3) | 7.100e-2(1.135e-2) |
| | $p$ | 3.020e-11 | 7.631e-9 | 1.303e-1 | 7.845e-1 | - |
| | $h$ | 1 | 1 | 0 | 0 | - |
| DF12 | (10,10) | **1.097e-2(6.747e-3)** | 4.197e-1(7.550e-2) | 5.162e-2(2.343e-3) | 7.576e-1(1.038e-1) | 6.442e-1(3.084e-2) |
| | $p$ | 3.020e-11 | 1.157e-7 | 7.551e-9 | 4.515e-2 | - |
| | $h$ | 1 | 1 | 1 | 1 | - |
| DF13 | (10,10) | **1.507e-1(3.759e-3)** | 5.447e-1(1.341e-1) | 3.777e-1(6.890e-2) | 2.267e+0(9.289e-1) | 9.128e-1(3.035e-2) |
| | $p$ | 2.433e-5 | 4.376e-1 | 5.517e-1 | 3.831e-5 | - |
| | $h$ | 1 | 0 | 0 | 1 | - |
| DF14 | (10,10) | **1.634e-1(4.688e-4)** | 4.563e-1(1.826e-1) | 1.097e-1(1.957e-3) | 3.328e-1(1.062e-1) | 6.349e-1(1.172e-2) |
| | $p$ | 3.020e-11 | 8.187e-1 | 2.657e-11 | 1.494e-1 | - |
| | $h$ | 1 | 0 | 1 | 0 | - |
| | ‡/†/ℓ | 5/8/1 | 5/3/6 | 5/6/3 | 9/1/4 | - |

**Table 9. Mean and standard deviation values of MSP obtained by five algorithms for $(n_t, \tau_t) = (10,20)$.**

| Fun. | $(n_t, \tau_t)$ | MOEA/D-FD | TrDMOEA | MoE | PPS | FBP |
|------|------|------|------|------|------|------|
| DF1 | (10,20) | 8.483e-3(1.011e-4) | 3.311e-2(1.079e-2) | **4.070e-3(6.737e-5)** | 6.887e-2(2.043e-2) | 7.835e-3(3.307e-3) |
| | $p$ | 2.433e-5 | 4.205e-2 | 3.040e-9 | 2.669e-9 | - |
| | $h$ | 1 | 1 | 1 | 1 | - |
| DF2 | (10,20) | 1.110e-2(2.392e-4) | 2.654e-2(2.991e-3) | **7.301e-3(2.132e-4)** | 7.160e-2(3.515e-2) | 1.987e-1(2.254e-2) |
| | $p$ | 3.020e-11 | 1.157e-7 | 3.564e-11 | 6.913e-4 | - |
| | $h$ | 1 | 1 | 1 | 1 | - |
| DF3 | (10,20) | **1.166e-2(4.839e-4)** | 4.869e-2(4.307e-2) | 3.348e-2(1.695e-4) | 3.768e-1(2.157e-1) | 2.138e-2(1.245e-2) |
| | $p$ | 6.765e-5 | 8.722e-7 | 7.571e-6 | 1.957e-10 | - |
| | $h$ | 1 | 1 | 1 | 1 | - |
| DF4 | (10,20) | 1.207e-1(8.165e-3) | 3.124e-1(1.519e-1) | **1.573e-2(2.944e-4)** | 2.843e+0(5.168e+0) | 9.384e-2(4.569e-2) |
| | $p$ | 1.108e-6 | 1.659e-1 | 8.105e-1 | 6.203e-4 | - |
| | $h$ | 1 | 0 | 0 | 1 | - |
| DF5 | (10,20) | 9.809e-3(1.202e-4) | 4.026e-2(1.164e-2) | 1.887e-2(3.624e-3) | 1.012e-1(5.111e-2) | **9.622e-3(3.506e-3)** |
| | $p$ | 6.912e-4 | 7.283e-1 | 5.650e-3 | 1.287e-9 | - |
| | $h$ | 1 | 0 | 1 | 1 | - |
| DF6 | (10,20) | **1.174e-1(2.998e-2)** | 4.886e+0(2.724e-1) | 2.613e-1(4.861e-2) | 4.765e+1(2.884e+1) | 1.955e+0(3.041e-1) |
| | $p$ | 1.777e-10 | 4.730e-6 | 1.662e-10 | 1.411e-9 | - |
| | $h$ | 1 | 1 | 1 | 1 | - |
| DF7 | (10,20) | 1.865e-2(1.285e-3) | 1.164e+3(1.147e+3) | 1.384e+1(1.939e+0) | 1.673e-2(1.982e-3) | **6.568e-3(2.962e-4)** |
| | $p$ | 3.020e-11 | 3.338e-11 | 3.267e-11 | 7.119e-9 | - |
| | $h$ | 1 | 1 | 1 | 1 | - |
| DF8 | (10,20) | 1.848e-2(1.100e-3) | 7.476e-2(3.271e-2) | 1.801e-2(3.692e-3) | 1.015e-1(5.671e-2) | **1.798e-2(7.534e-3)** |
| | $p$ | 4.675e-2 | 7.197e-5 | 6.581e-1 | 4.311e-8 | - |
| | $h$ | 1 | 1 | 0 | 1 | - |
| DF9 | (10,20) | **1.054e-2(8.937e-4)** | 2.816e-1(8.860e-2) | 3.886e-2(6.564e-3) | 3.955e-1(1.687e-1) | 5.778e-2(3.320e-3) |
| | $p$ | 7.389e-11 | 1.453e-1 | 3.627e-2 | 3.778e-2 | - |
| | $h$ | 1 | 0 | 1 | 1 | - |
| DF10 | (10,20) | **4.028e-2(1.551e-2)** | 3.042e-1(1.883e-2) | 2.762e-2(1.463e-3) | 1.060e+0(2.206e-1) | 1.059e+0(8.679e-2) |
| | $p$ | 3.020e-11 | 4.616e-10 | 2.374e-10 | 1.260e-1 | - |
| | $h$ | 1 | 1 | 1 | 0 | - |
| DF11 | (10,20) | **2.347e-2(4.776e-4)** | 6.834e-2(1.956e-3) | 6.086e-2(6.000e-4) | 4.900e-2(2.209e-3) | 5.014e-2(2.631e-3) |
| | $p$ | 3.020e-11 | 1.330e-10 | 2.005e-10 | 7.618e-1 | - |
| | $h$ | 1 | 1 | 1 | 0 | - |
| DF12 | (10,20) | **1.878e-2(1.032e-2)** | 2.425e-1(5.327e-2) | 5.230e-2(2.110e-3) | 7.050e-1(1.003e-1) | 6.001e-1(2.135e-2) |
| | $p$ | 3.020e-11 | 9.919e-11 | 8.601e-9 | 4.841e-2 | - |
| | $h$ | 1 | 1 | 1 | 1 | - |
| DF13 | (10,20) | **1.661e-1(3.742e-3)** | 4.874e-1(4.391e-2) | 3.324e-1(5.423e-2) | 1.110e+0(3.715e-1) | 2.025e-1(1.381e-2) |
| | $p$ | 5.493e-1 | 5.462e-9 | 6.881e-1 | 2.610e-10 | - |
| | $h$ | 0 | 1 | 0 | 1 | - |
| DF14 | (10,20) | **1.819e-2(5.114e-4)** | 2.086e-1(4.616e-2) | 1.014e-1(9.106e-4) | 1.491e-1(3.352e-2) | 3.882e-1(1.177e-2) |
| | $p$ | 3.020e-11 | 1.114e-3 | 4.518e-4 | 1.337e-5 | - |
| | $h$ | 1 | 1 | 1 | 1 | - |
| | ‡/†/ℓ | 5/9/0 | 7/4/3 | 4/7/3 | 10/2/2 | - |

of finding high-quality approximations to the POF, as shown by the large MIGD and MHV results recorded in Tables 1–9, respectively. Overall, FBP seems less sensitive to the frequency and severity of change, as can be observed from its gradual improvement on the three measures when $\tau_t$ and $n_t$ increase in most cases, for which the compared algorithms have drastic changes in their performance.

Fig 3 presents some convergence graphs of the mean IGD values for a majority of the benchmark functions. It is obvious that FBP shows more stable ability and recovers faster from dynamic changes in most case, thereby gaining higher convergence process compared with the others. For DF10, FBP does not perform well for the first a few environments, but it has significant advantage over its peers in later environments. The overall performance of FBP is better than the others on DF8.

Figs 4–7 plot some POF approximation on DF3, DF5, DF7 and DF8, which are intuitive representations of the solutions. It is obvious that FBP performs better than the compared

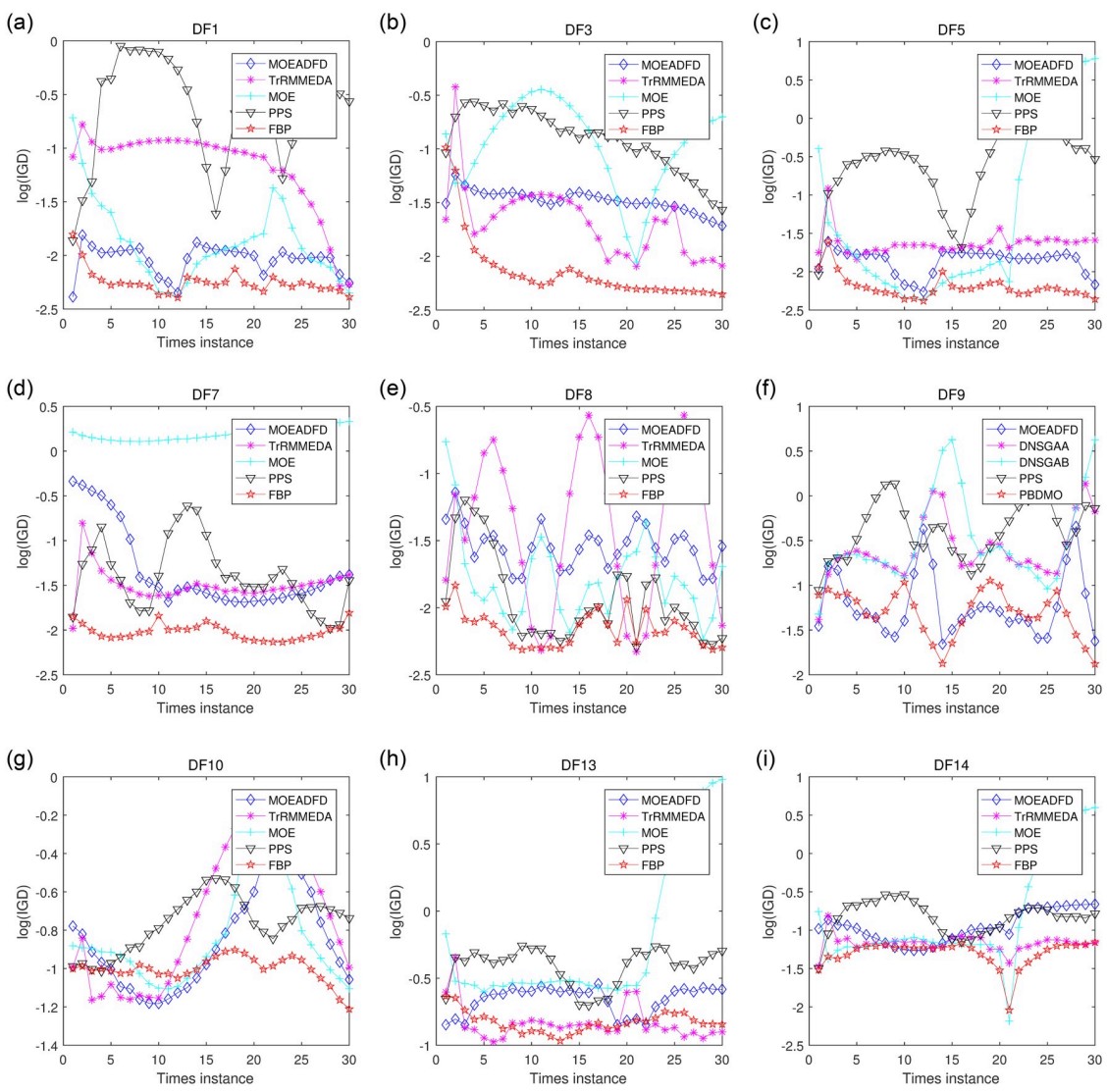

**Fig 3. Mean IGD curves for different problems with $n_t = 10$ and $\tau_t = 1$.**

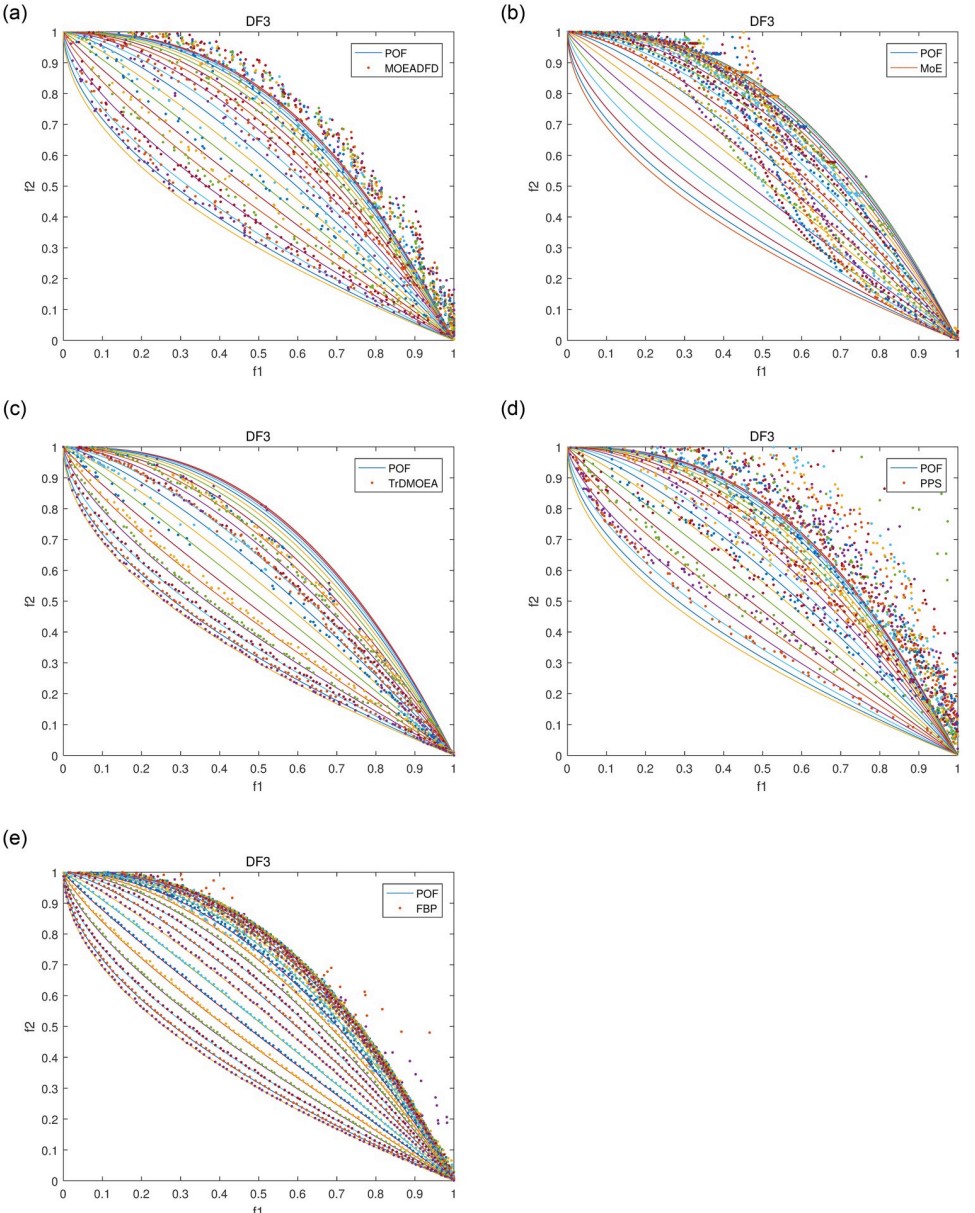

**Fig 4. POF approximations of five algorithms for DF3 with $n_t$ = 10 and $\tau_t$ = 10.**

algorithms. The approximations demonstrate clearly that FBP has excellent tracking ability in varying environments, but it may generate some boundary individuals in DF8.

Apart from the above analysis, to investigate the performance of the proposed dynamic dynamic multiobjective algorithm further, some recent MO algorithms (MOGOA, MOMVO, MOALO and MSSA) are employed for comparisons. They are equipped with the same reaction mechanism used in FBP, Tables 10–12 record the simulation results including mean values, standard deviation and t-test values. It can be seen that FBP outperforms the compared algorithms on the majority of test problems based on MIGD and MHV results, and the *p*-values summarized in the bottom of Tables also indicate that the differences among them are significant. For the MSP, the advantages of the algorithm are not obvious on the three functions

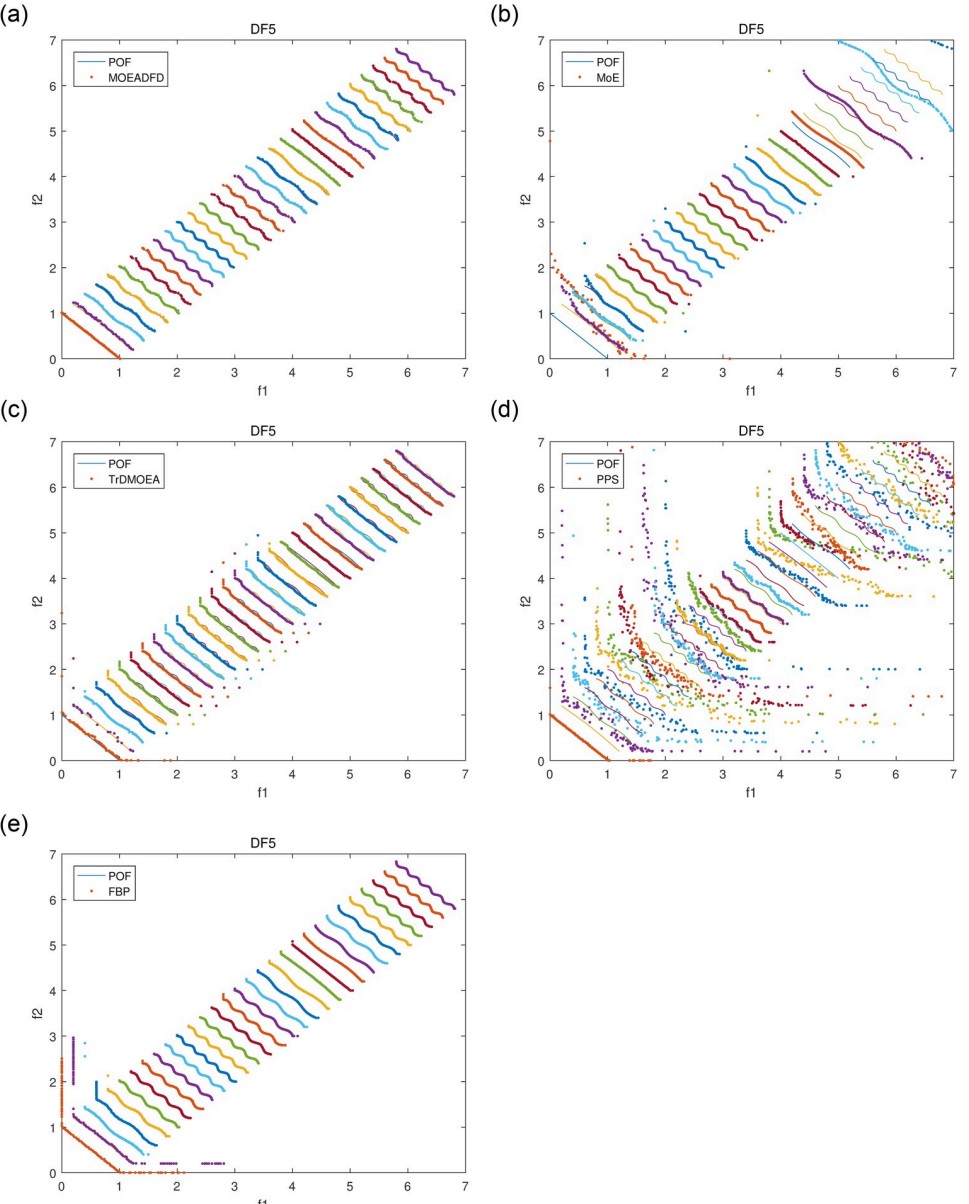

**Fig 5. POF approximations of five algorithms for DF5 with $n_t = 10$ and $\tau_t = 10$.**

(DF2, DF11 and DF13), but the *p*-values show that the differences among them are not significant. Totally, FBP is able to generate competitive results with respective to other compared approaches.

## 5 Discussion

### 5.1 Component analysis

As mentioned before, the proposed strategy contains three different key components. This subsection aims to discuss the role that each component plays in dealing with dynamic environment. Specifically, to demonstrate the importance of the linear prediction model with two

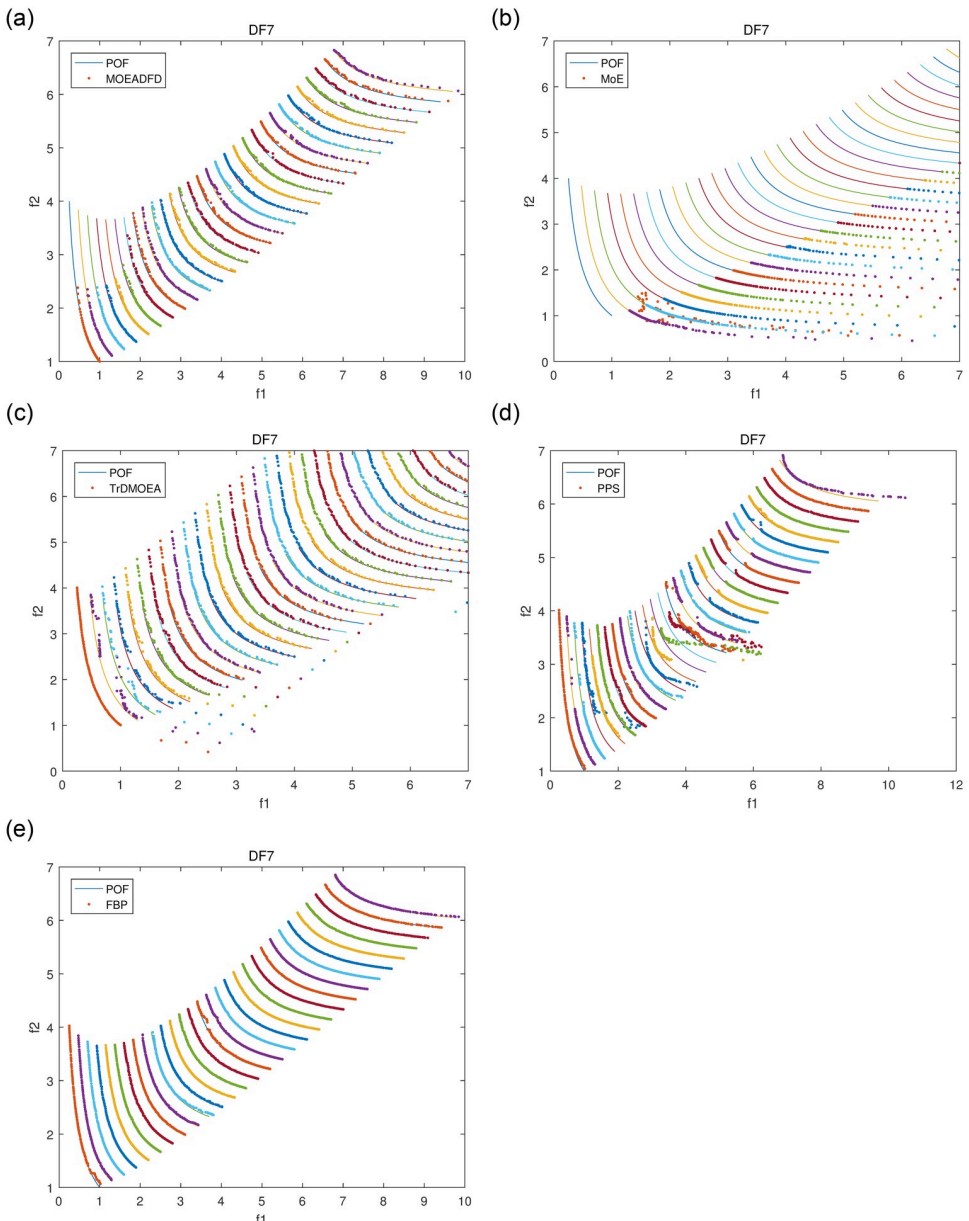

**Fig 6. POF approximations of five algorithms for DF7 with $n_t$ = 10 and $\tau_t$ = 10.**

different stepsizes, a one step prediction model is utilized to replace the proposed two steps strategy for predicting non-dominated solutions. This is, the step value is set to one ($step$ = 1), which is a common setting in most existing prediction-based techniques, and the variant is named FBPV1. To demonstrate that the fitting-based strategy has important effect on the proposed strategy, FBPV2 is designed by removing the sampling strategy; in the other words, FBPV2 just has two prediction strategies. Similarly, to study the role of the third strategy, FBP is also modified by excluding the reference sampling strategy, called FBPV3.

These three variants are compared with the original FBP, and Table 13 report the corresponding computing results. The following discusses the influence of each component in detail.

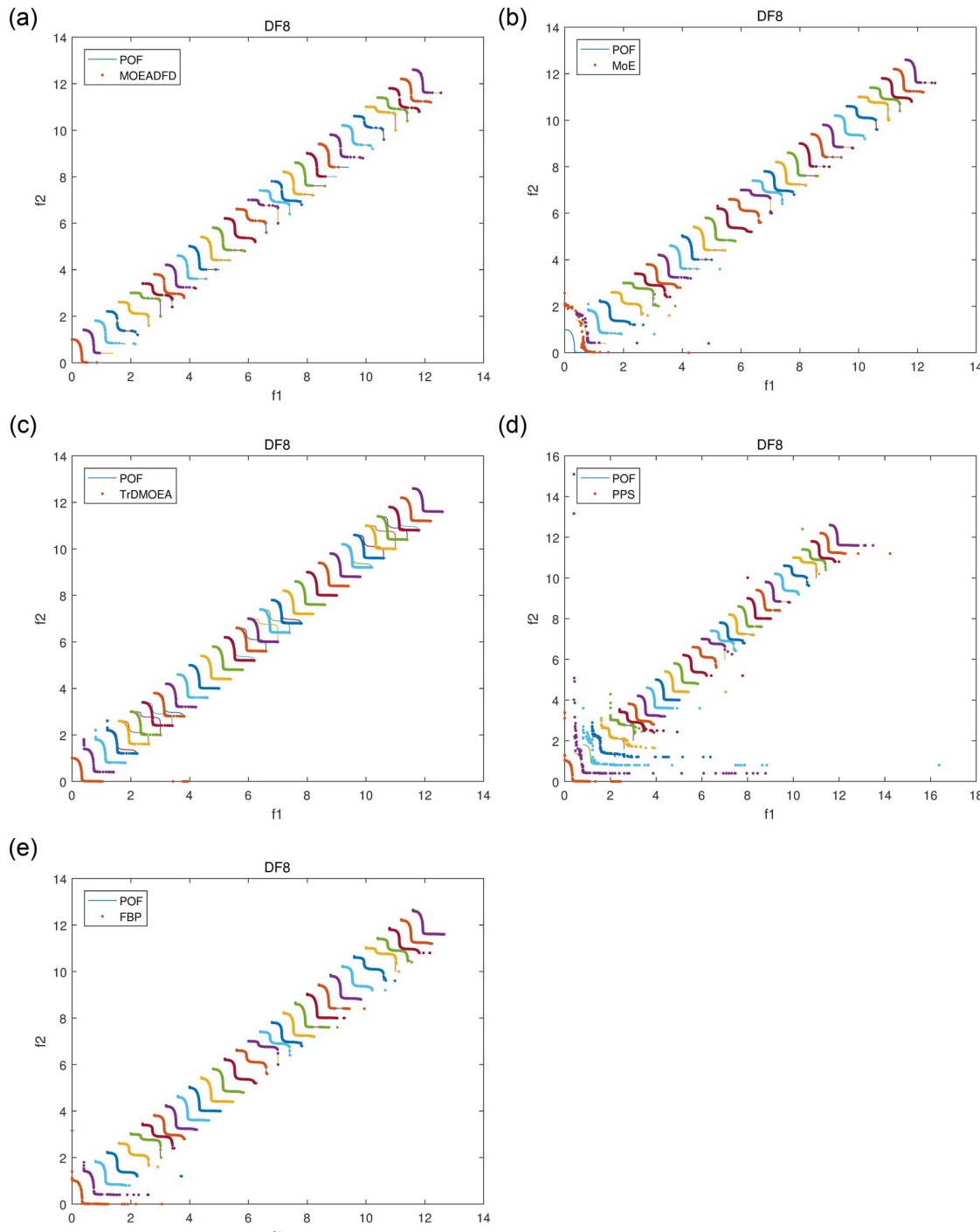

**Fig 7. POF approximations of five algorithms for DF8 with $n_t = 10$ and $\tau_t = 10$.**

**5.1.1 Linear prediction mode.** It is clear that FBP is much superior to FBPV1 in terms of MIGD on some cases, but the differences among them are not too significant in most of test problems based on the $p$-values. The reason may come from the fact that FBP utilizes a two-step based prediction strategy, which would generate more boundary individuals than FBPV1.

**Table 10. Performance comparison of different multiobjective algorithms variants on MIGD.**

| Fun. | $(n_t, \tau_t)$ | MOGOA | MOMVO | MOALO | MSSA | FBP |
|------|------|------|------|------|------|------|
| DF1 | (10,10) | 4.738e-2(8.374e-3) | 1.186e-2(3.281e-4) | 4.146e-2(7.050e-3) | 3.989e-2(4.897e-3) | **5.844e-3(2.682e-4)** |
| | $p$ | 2.670e-9 | 4.182e-9 | 7.043e-7 | 6.046e-7 | - |
| | $h$ | 1 | 1 | 1 | 1 | - |
| DF2 | (10,10) | 1.903e-1(1.041e-2) | 1.193e-1(6.074e-3) | 1.946e-1(7.243e-3) | 1.535e-1(6.676e-3) | **3.937e-2(4.765e-3)** |
| | $p$ | 5.895e-1 | 1.174e-3 | 5.895e-1 | 9.646e-2 | - |
| | $h$ | 0 | 1 | 0 | 0 | - |
| DF3 | (10,10) | 7.898e-2(1.321e-2) | 5.270e-2(8.431e-3) | 7.648e-2(1.519e-2) | 6.577e-2(1.529e-2) | **1.162e-2(5.596e-3)** |
| | $p$ | 1.911e-2 | 4.637e-3 | 6.972e-3 | 2.709e-2 | - |
| | $h$ | 1 | 1 | 1 | 1 | - |
| DF4 | (10,10) | 1.952e-1(2.347e-2) | 9.334e-2(5.608e-3) | 2.993e-1(4.554e-2) | 1.359e-1(1.584e-2) | **7.029e-2(2.230e-3)** |
| | $p$ | 5.692e-1 | 9.470e-1 | 4.119e-1 | 8.073e-1 | - |
| | $h$ | 0 | 0 | 0 | 0 | - |
| DF5 | (10,10) | 3.651e-2(2.985e-2) | 1.126e-2(6.515e-4) | 4.380e-2(4.928e-3) | 2.163e-2(1.498e-3) | **6.832e-3(7.967e-4)** |
| | $p$ | 3.945e-8 | 2.154e-10 | 1.873e-7 | 5.967e-9 | - |
| | $h$ | 1 | 1 | 1 | 1 | - |
| DF6 | (10,10) | 1.812e+0(3.526e-1) | 6.598e-1(2.662e-2) | 1.419e+0(1.912e-1) | 1.482e+0(2.395e-1) | **6.140e-1(4.412e-2)** |
| | $p$ | 1.787e-6 | 4.532e-8 | 5.135e-5 | 2.095e-7 | - |
| | $h$ | 1 | 1 | 1 | 1 | - |
| DF7 | (10,10) | 5.798e-2(6.698e-3) | 5.919e-2(1.181e-2) | 5.541e-2(6.659e-3) | 6.647e-2(4.721e-3) | **9.785e-3(4.370e-4)** |
| | $p$ | 7.394e-1 | 4.290e-1 | 4.643e-1 | 7.845e-1 | - |
| | $h$ | 0 | 0 | 0 | 0 | - |
| DF8 | (10,10) | 2.617e-2(2.368e-3) | 1.378e-2(1.528e-3) | 2.503e-2(2.139e-3) | 1.835e-2(2.784e-3) | **7.152e-3(5.012e-4)** |
| | $p$ | 5.895e-1 | 5.011e-1 | 6.414e-1 | 8.418e-1 | - |
| | $h$ | 0 | 0 | 0 | 0 | - |
| DF9 | (10,10) | 1.879e-1(1.085e-2) | 1.530e-1(4.694e-3) | 1.871e-1(6.347e-3) | 1.391e-1(6.706e-3) | **6.755e-2(5.406e-3)** |
| | $p$ | 2.133e-5 | 6.528e-8 | 2.773e-5 | 2.028e-7 | - |
| | $h$ | 1 | 1 | 1 | 1 | - |
| DF10 | (10,10) | 1.151e-1(8.943e-3) | 8.616e-2(4.964e-3) | 1.215e-1(7.780e-3) | 1.117e-1(9.539e-3) | **1.001e-1(3.921e-3)** |
| | $p$ | 1.628e-2 | 2.068e-2 | 2.678e-6 | 2.755e-3 | - |
| | $h$ | 1 | 1 | 1 | 1 | - |
| DF11 | (10,10) | 6.833e-1(1.025e-2) | 6.449e-1(2.818e-3) | 6.567e-1(4.076e-3) | 6.688e-1(3.715e-3) | **6.436e-1(1.138e-3)** |
| | $p$ | 5.395e-1 | 9.234e-1 | 9.234e-1 | 7.394e-1 | - |
| | $h$ | 0 | 0 | 0 | 0 | - |
| DF12 | (10,10) | 4.252e-1(4.889e-2) | 5.188e-1(5.700e-2) | 5.616e-1(6.027e-2) | 5.951e-1(4.716e-2) | **2.831e-1(1.074e-2)** |
| | $p$ | 3.831e-5 | 7.773e-9 | 2.921e-9 | 6.518e-9 | - |
| | $h$ | 1 | 1 | 1 | 1 | - |
| DF13 | (10,10) | 2.004e-1(9.503e-3) | 1.097e-1(6.769e-3) | 2.290e-1(4.475e-2) | 1.828e-1(2.534e-2) | **1.500e-1(4.638e-3)** |
| | $p$ | 3.770e-4 | 5.533e-8 | 2.015e-8 | 1.766e-3 | - |
| | $h$ | 1 | 1 | 1 | 1 | - |
| DF14 | (10,10) | 6.768e-2(2.941e-3) | 4.598e-2(1.876e-3) | 6.818e-2(4.887e-3) | 5.744e-2(4.825e-3) | **5.251e-2(1.619e-3)** |
| | $p$ | 7.727e-2 | 6.145e-2 | 1.188e-1 | 1.537e-1 | - |
| | $h$ | 1 | 0 | 0 | 0 | - |
| | ‡/†/ℓ | 9/0/5 | 8/2/4 | 8/0/6 | 8/0/6 | - |

**Table 11. Performance comparison of different multiobjective algorithms variants on MHV.**

| Fun. | $(n_t, \tau_t)$ | MOGOA | MOMVO | MOALO | MSSA | FBP |
|------|------|-------|-------|-------|------|-----|
| DF1 | (10,10) | 1.555e+0(1.419e-2) | 1.648e+0(1.857e-3) | 1.558e+0(2.210e-2) | 1.559e+0(1.638e-2) | **1.668e+0(9.101e-4)** |
| | $p$ | 6.046e-7 | 7.088e-8 | 1.411e-9 | 1.429e-8 | - |
| | $h$ | 1 | 1 | 1 | 1 | - |
| DF2 | (10,10) | 1.456e+0(2.049e-2) | 1.695e+0(9.909e-3) | 1.452e+0(1.791e-2) | 1.546e+0(1.385e-2) | **1.798e+0(1.307e-2)** |
| | $p$ | 6.046e-7 | 7.088e-8 | 1.411e-9 | 1.429e-8 | - |
| | $h$ | 1 | 1 | 1 | 1 | - |
| DF3 | (10,10) | 1.443e+0(2.120e-2) | 1.491e+0(1.724e-2) | 1.448e+0(3.963e-2) | 1.453e+0(4.203e-2) | **1.614e+0(1.295e-2)** |
| | $p$ | 6.046e-7 | 7.088e-8 | 1.411e-9 | 1.429e-8 | - |
| | $h$ | 1 | 1 | 1 | 1 | - |
| DF4 | (10,10) | 6.785e+0(1.086e-1) | 7.280e+0(2.941e-2) | 6.512e+0(5.560e-2) | 7.098e+0(6.747e-2) | **7.588e+0(1.027e-2)** |
| | $p$ | 6.046e-7 | 7.088e-8 | 1.411e-9 | 1.429e-8 | - |
| | $h$ | 1 | 1 | 1 | 1 | - |
| DF5 | (10,10) | 1.660e+0(8.226e-3) | 1.727e+0(1.175e-3) | 1.646e+0(1.403e-2) | 1.691e+0(4.786e-3) | **1.734e+0(1.575e-3)** |
| | $p$ | 6.046e-7 | 7.088e-8 | 1.411e-9 | 1.429e-8 | - |
| | $h$ | 1 | 1 | 1 | 1 | - |
| DF6 | (10,10) | 1.065e+0(2.146e-2) | **1.152e+0(1.215e-2)** | 1.109e+0(1.326e-2) | 1.151e+0(3.038e-2) | 1.214e-1(3.713e-2) |
| | $p$ | 6.046e-7 | 7.088e-8 | 1.411e-9 | 1.429e-8 | - |
| | $h$ | 1 | 1 | 1 | 1 | - |
| DF7 | (10,10) | 3.312e+0(2.168e-2) | 3.316e+0(1.895e-2) | 3.310e+0(2.595e-2) | 3.317e+0(1.804e-2) | **3.465e+0(2.298e-3)** |
| | $p$ | 6.046e-7 | 7.088e-8 | 1.411e-9 | 1.429e-8 | - |
| | $h$ | 1 | 1 | 1 | 1 | - |
| DF8 | (10,10) | 1.739e+0(1.104e-2) | 1.780e+0(3.679e-3) | 1.743e+0(6.805e-3) | 1.756e+0(9.661e-3) | **1.784e+0(1.653e-3)** |
| | $p$ | 6.046e-7 | 7.088e-8 | 1.411e-9 | 1.429e-8 | |
| | $h$ | 1 | 1 | 1 | 1 | - |
| DF9 | (10,10) | 1.290e+0(1.727e-2) | 1.394e+0(1.317e-2) | 1.281e+0(1.638e-2) | 1.374e+0(1.895e-2) | **1.571e+0(1.172e-2)** |
| | $p$ | 6.046e-7 | 7.088e-8 | 1.411e-9 | 1.429e-8 | - |
| | $h$ | 1 | 1 | 1 | 1 | - |
| DF10 | (10,10) | 1.086e+0(3.402e-2) | 1.213e+0(4.513e-2) | 1.039e+0(2.902e-2) | 1.116e+0(4.084e-2) | **1.379e+0(1.174e-2)** |
| | $p$ | 6.528e-8 | 3.564e-4 | 2.227e-9 | 1.028e-6 | - |
| | $h$ | 1 | 1 | 1 | 1 | - |
| DF11 | (10,10) | 2.741e-1(1.331e-2) | 3.646e-1(2.381e-3) | 2.889e-1(2.340e-2) | 2.854e-1(8.379e-3) | **3.714e-1(2.397e-3)** |
| | $p$ | 4.464e-1 | 6.375e-1 | 5.106e-1 | 6.414e-1 | - |
| | $h$ | 0 | 0 | 0 | 0 | - |
| DF12 | (10,10) | 3.194e+0(8.433e-2) | **3.468e+0(5.777e-2)** | 3.111e+0(8.543e-2) | 3.428e+0(4.491e-2) | 3.458e+0(1.013e-2) |
| | $p$ | 4.353e-5 | 2.226e-1 | 1.996e-5 | 4.033e-3 | - |
| | $h$ | 1 | 0 | 1 | 1 | - |
| DF13 | (10,10) | 6.611e+0(1.151e-1) | **7.381e+0(1.326e-2)** | 6.358e+0(2.959e-1) | 6.702e+0(1.776e-1) | **7.105e+0(2.803e-2)** |
| | $p$ | 5.804e-3 | 5.746e-2 | 2.839e-4 | 3.644e-2 | - |
| | $h$ | 1 | 0 | 1 | 1 | - |
| DF14 | (10,10) | 9.861e-1(1.291e-2) | 1.080e+0(5.046e-3) | 9.662e-1(2.176e-2) | **1.011e+0(1.016e-2)** | 1.073e+0(1.995e-3) |
| | $p$ | 3.329e-1 | 8.187e-1 | 2.226e-1 | 4.553e-1 | - |
| | $h$ | 0 | 0 | 0 | 0 | - |
| | ‡/†/ℓ | 11/1/2 | 9/1/4 | 11/1/2 | 11/1/2 | - |

**Table 12. Performance comparison of different multiobjective algorithms variants on MSP.**

| Fun. | $(n_t, \tau_t)$ | MOGOA | MOMVO | MOALO | MSSA | FBP |
|------|------|-------|-------|-------|------|-----|
| DF1 | (10,10) | 2.119e-2(4.378e-3) | 8.133e-3(4.404e-4) | 2.244e-2(3.494e-3) | 1.618e-2(2.992e-3) | **5.810e-3(1.984e-3)** |
| | $p$ | 9.833e-8 | 3.020e-11 | 1.473e-7 | 2.921e-9 | - |
| | $h$ | 1 | 1 | 1 | 1 | - |
| DF2 | (10,10) | 1.032e-1(2.179e-2) | **1.971e-2(1.688e-3)** | 8.844e-2(1.769e-2) | 1.478e-1(1.531e-2) | 1.309e-1(2.166e-2) |
| | $p$ | 1.260e-1 | 4.311e-8 | 2.838e-1 | 7.727e-2 | - |
| | $h$ | 0 | 1 | 0 | 0 | - |
| DF3 | (10,10) | 6.572e-2(2.488e-2) | 2.424e-2(5.338e-3) | 3.950e-2(1.503e-2) | 5.305e-2(1.778e-2) | **1.426e-2(7.014e-3)** |
| | $p$ | 1.202e-8 | 3.020e-11 | 1.777e-10 | 2.439e-9 | - |
| | $h$ | 1 | 1 | 1 | 1 | - |
| DF4 | (10,10) | 3.374e-1(9.633e-2) | **5.516e-2(1.084e-2)** | 2.675e-1(1.511e-1) | 5.777e-1(2.361e-1) | 7.809e-2(3.562e-2) |
| | $p$ | 6.669e-3 | 8.073e-1 | 1.628e-2 | 1.597e-3 | - |
| | $h$ | 1 | 0 | 1 | 1 | - |
| DF5 | (10,10) | 3.609e-2(9.195e-3) | **7.980e-3(1.189e-3)** | 4.291e-2(1.395e-2) | 3.495e-2(8.841e-3) | 9.507e-3(3.978e-3) |
| | $p$ | 2.195e-8 | 4.077e-11 | 1.873e-7 | 3.081e-8 | - |
| | $h$ | 1 | 1 | 1 | 1 | - |
| DF6 | (10,10) | 4.018e+0(7.480e-1) | **1.507e+0(3.991e-1)** | 7.344e+0(1.046e+1) | 3.918e+0(8.249e-1) | 9.134e+0(4.567e-1) |
| | $p$ | 1.729e-7 | 1.411e-9 | 5.600e-7 | 1.596e-7 | - |
| | $h$ | 1 | 1 | 1 | 1 | - |
| DF7 | (10,10) | 3.234e-2(4.062e-3) | 3.989e-2(4.618e-3) | 3.097e-2(3.178e-3) | 3.214e-2(5.375e-3) | **7.677e-3(2.650e-4)** |
| | $p$ | 1.325e-4 | 5.092e-8 | 8.292e-6 | 2.499e-3 | - |
| | $h$ | 1 | 1 | 1 | 1 | - |
| DF8 | (10,10) | 4.724e-2(1.823e-2) | 2.187e-2(2.674e-3) | 3.871e-2(7.640e-3) | 3.894e-2(1.280e-2) | **1.933e-2(4.811e-3)** |
| | $p$ | 6.353e-2 | 1.407e-4 | 2.709e-2 | 1.911e-2 | - |
| | $h$ | 0 | 1 | 1 | 1 | - |
| DF9 | (10,10) | 2.794e-1(7.429e-2) | 1.975e-1(2.446e-2) | 2.361e-1(3.913e-2) | 2.266e-1(8.619e-2) | **1.893e-1(3.860e-2)** |
| | $p$ | 1.597e-3 | 5.462e-9 | 1.585e-4 | 1.493e-4 | - |
| | $h$ | 1 | 1 | 1 | 1 | - |
| DF10 | (10,10) | 3.483e-1(6.588e-2) | **2.552e-1(5.670e-2)** | 3.458e-1(4.913e-2) | 3.047e-1(6.561e-2) | 9.422e-1(1.463e-2) |
| | $p$ | 1.287e-9 | 4.616e-10 | 4.573e-9 | 2.034e-9 | - |
| | $h$ | 1 | 1 | 1 | 1 | - |
| DF11 | (10,10) | 6.543e-2(3.525e-3) | 5.381e-2(2.150e-3) | **5.371e-2(6.281e-3)** | 6.748e-2(2.277e-3) | 7.100e-2(1.135e-2) |
| | $p$ | 5.592e-1 | 3.848e-3 | 1.413e-1 | 2.905e-1 | - |
| | $h$ | 0 | 1 | 0 | 0 | - |
| DF12 | (10,10) | 2.577e-1(4.956e-2) | **1.323e-1(4.000e-2)** | 1.914e-1(3.848e-2) | 1.788e-1(3.000e-2) | 6.442e-1(3.084e-2) |
| | $p$ | 3.474e-10 | 3.120e-11 | 1.464e-10 | 2.610e-10 | - |
| | $h$ | 1 | 1 | 1 | 1 | - |
| DF13 | (10,10) | 3.758e-1(1.384e-1) | **1.196e-1(2.152e-2)** | 3.075e-1(8.518e-2) | 3.372e-1(7.603e-2) | 9.128e-1(3.035e-2) |
| | $p$ | 3.632e-1 | 5.462e-9 | 3.790e-1 | 9.626e-2 | - |
| | $h$ | 0 | 1 | 0 | 0 | - |
| DF14 | (10,10) | 1.214e-1(4.976e-2) | 4.191e-2(1.127e-2) | **1.188e-2(4.057e-2)** | 9.399e-2(4.535e-2) | 6.349e-1(1.172e-2) |
| | $p$ | 7.088e-8 | 3.690e-11 | 5.600e-7 | 6.528e-8 | - |
| | $h$ | 1 | 1 | 1 | 1 | - |
| | ‡/†/≀ | 6/4/4 | 5/8/1 | 7/4/3 | 7/4/3 | - |

**Table 13. Performance comparison of different FBP variants on MIGD.**

| Fun. | $(n_t, \tau_t)$ | FBPV1 | FBPV2 | FBPV3 | FBP |
|------|------|------|------|------|------|
| DF1 | (10,10) | 6.046e-3(2.750e-4) | 5.934e-3(2.898e-4) | 1.652e-2(6.447e-3) | **5.844e-3(2.682e-4)** |
| | $p$ | 5.011e-1 | 1.260e-1 | 1.236e-3 | - |
| | $h$ | 0 | 0 | 1 | - |
| DF2 | (10,10) | **3.844e-2(3.655e-3)** | 5.561e-2(5.542e-3) | 3.877e-2(4.215e-3) | 3.937e-2(4.765e-3) |
| | $p$ | 9.352e-1 | 9.334e-2 | 9.941e-1 | - |
| | $h$ | 0 | 0 | 0 | - |
| DF3 | (10,10) | 1.472e-2(9.277e-3) | 2.610e-2(2.915e-2) | 1.434e-2(1.053e-3) | **1.162e-2(5.596e-3)** |
| | $p$ | 6.952e-1 | 2.643e-1 | 6.843e-1 | - |
| | $h$ | 0 | 0 | 0 | - |
| DF4 | (10,10) | 7.058e-2(2.557e-3) | 7.213e-2(2.659e-3) | 7.247e-2(3.550e-3) | **7.029e-2(2.230e-3)** |
| | $p$ | 8.303e-1 | 7.731e-1 | 9.234e-1 | - |
| | $h$ | 0 | 0 | 0 | - |
| DF5 | (10,10) | 6.956e-3(5.116e-4) | 7.040e-3(1.403e-2) | 1.336e-2(3.222e-3) | **6.832e-3(7.967e-4)** |
| | $p$ | 6.520e-1 | 4.204e-1 | 2.282e-1 | - |
| | $h$ | 0 | 0 | 0 | - |
| DF6 | (10,10) | 6.248e-1(5.066e-2) | 6.952e-1(7.643e-2) | 1.831e+0(3.855e-1) | **6.140e-1(4.412e-2)** |
| | $p$ | 9.117e-1 | 4.119e-1 | 1.335e-1 | - |
| | $h$ | 0 | 0 | 0 | - |
| DF7 | (10,10) | 9.643e-3(2.833e-4) | **9.564e-3(2.868e-4)** | 1.375e-2(2.474e-3) | 9.785e-3(4.370e-4) |
| | $p$ | 7.618e-1 | 4.825e-1 | 2.009e-1 | - |
| | $h$ | 0 | 0 | 0 | - |
| DF8 | (10,10) | 7.188e-3(5.881e-4) | 7.190e-3(4.454e-4) | 8.510e-3(4.286e-4) | **7.152e-3(5.012e-4)** |
| | $p$ | 7.506e-1 | 6.952e-1 | 3.329e-1 | - |
| | $h$ | 0 | 0 | 0 | - |
| DF9 | (10,10) | 7.293e-2(6.506e-3) | 7.189e-2(8.008e-3) | 7.093e-2(7.740e-3) | **6.755e-2(5.406e-3)** |
| | $p$ | 4.733e-1 | 6.627e-1 | 7.062e-1 | - |
| | $h$ | 0 | 0 | 0 | - |
| DF10 | (10,10) | 1.119e-1(7.538e-3) | 1.095e-1(5.610e-3) | 1.027e-1(5.233e-3) | **1.001e-1(3.921e-3)** |
| | $p$ | 3.020e-11 | 3.020e-11 | 3.020e-11 | - |
| | $h$ | 1 | 1 | 1 | - |
| DF11 | (10,10) | 6.439e-1(1.579e-3) | 6.441e-1(1.666e-3) | 6.458e-1(1.975e-3) | **6.436e-1(1.138e-3)** |
| | $p$ | 4.975e-11 | 4.972e-11 | 4.077e-11 | - |
| | $h$ | 1 | 1 | 1 | - |
| DF12 | (10,10) | 2.947e-1(6.804e-3) | 2.893e-1(7.764e-3) | 3.157e+2(1.578e+2) | **2.831e-1(1.074e-2)** |
| | $p$ | 3.338e-11 | 3.020e-11 | 3.020e-11 | - |
| | $h$ | 1 | 1 | 1 | - |
| DF13 | (10,10) | 1.800e-1(1.115e-2) | 1.529e-1(4.838e-3) | 1.660e-1(8.410e-3) | **1.500e-1(4.638e-3)** |
| | $p$ | 3.081e-8 | 5.462e-6 | 8.197e-7 | - |
| | $h$ | 1 | 1 | 1 | - |
| DF14 | (10,10) | 6.105e-2(5.593e-3) | 5.288e-2(1.826e-3) | 5.229e-2(1.697e-3) | **5.251e-2(1.619e-3)** |
| | $p$ | 4.504e-11 | 4.505e-11 | 6.066e-11 | - |
| | $h$ | 1 | 1 | 1 | - |
| | ‡/†/ℓ | 5/0/9 | 5/0/9 | 5/1/8 | - |

Therefore, the population diversity can be affected by too much non-dominated boundary solutions immediately. Despite that, the overall performance of the two-step technique performs much better than one-step strategy for the majority of the benchmark functions.

**5.1.2 Curve fitting-based strategy.**   It is not difficult to observe from the results that FBP outperforms the modified variant FBPV2 on most of the test functions. This means that the curve fitting-based strategy indeed helps improve the quality of population in varying environments. The reason may originate from the fact that the curve fitting-based strategy is designed by considering interlinks between variables, which helps to generate promising solutions to some extent.

The comparison between the three different variants and the proposed FBP illustrates that each part has an significant effect on the performance of FBP, and removing any of them reduces performance. Therefore, it is important to combine them together as in the FBP strategy.

**5.1.3 Sampling strategy.**   All the results illustrate that FBP performs much better than FBPV2 for almost all test problems, although FBP is slightly weaker than FBPV3 for DF14 problem. Thus, the designed sampling technique is able to improve the search ability of population in each varying environment clearly and can further improve the effectiveness of the proposed dynamic multiobjective optimization algorithm.

## 5.2 Influence of *step* values

As described before, the linear prediction model employs two different stepsizes, which are set to 1 and 0.3 for predicting non-dominated solutions, respectively. Here, to study whether the *step* values are well configured, *step* = 1 is fixed as it has proven effective in many prediction algorithms, and the other *step* is set to an increment of 0.2 from 0.1 to 0.7 (FBPS1-FBPS3). Numerical results in Table 14 for the fourteen functions shows that the algorithms become ineffective when *step* is too large shown by t-test values. The results illustrate that FBP outperforms other three versions on a majority of functions, although the differences between us are not very large on some cases. Therefore, it can be concluded from the experiment that FBP should utilize two different stepsize values (1 and 0.3) reasonably.

## 5.3 Influence of degree of polynomial regression

As a importance part of FBP, the curve fitting-based strategy has a significant parameter, the degree of polynomial regression (*dpf*). Here, the *dpf* is set to different values, with an increment of 1, from 1 to 4 (FBPL1-FBPL3) for exploring its influence on algorithms' performance. The comparison results recorded in Table 15 show that the proposed technique is superior to the other versions on almost all the test problems. Although the higher the degree, the better the goodness of fit, too high degree may result in over-fitting. Thus, it is important to properly select the degree of polynomial regression and the experimental analysis supports the decision made to choose a degree of two.

## 5.4 Influence of *cr* values

In the third strategy, the new prediction fitting curve is obtained based on Eqs (6) and (7). After that, it will be used to generate new individuals using (Eq 8), which involves two important parameters, the compression ratio (*cr*) and subpopulation (*Subpop*$_2$) size. The former is discussed in this subsection, and the latter will be analyzed below. *cr* ranges from 0.1 to 0.7, with an increment of 0.2 (FBPR1-FBPR3), and the results are summarized in Table 16. It is obvious that the original variant performs much better than the other three versions in almost

**Table 14. Performance comparison of FBP variants on MIGD for $(n_t, \tau_t) = (10,10)$.**

| Fun. | $(n_t, \tau_t)$ | FBPS1 | FBPS2 | FBPS3 | FBP |
|---|---|---|---|---|---|
| DF1 | (10,10) | 5.861e-3(3.024e-4) | 5.939e-3(3.137e-4) | 5.799e-3(3.571e-4) | **5.844e-3(2.682e-4)** |
|  | $p$ | 5.895e-1 | 8.187e-1 | 5.106e-1 | - |
|  | $h$ | 0 | 0 | 0 | - |
| DF2 | (10,10) | **3.695e-2(3.258e-3)** | 4.036e-2(4.304e-3) | 4.084e-2(3.900e-3) | 3.937e-2(4.765e-3) |
|  | $p$ | 8.766e-1 | 9.000e-1 | 6.843e-1 | - |
|  | $h$ | 0 | 0 | 0 | - |
| DF3 | (10,10) | 1.497e-2(9.258e-3) | 1.236e-2(7.067e-3) | 1.265e-2(5.964e-3) | **1.162e-2(5.596e-3)** |
|  | $p$ | 7.172e-1 | 8.187e-1 | 8.650e-1 | - |
|  | $h$ | 0 | 0 | 0 | - |
| DF4 | (10,10) | 7.054e-2(2.689e-3) | 7.184e-2(3.091e-3) | 7.183e-2(3.456e-3) | **7.029e-2(2.230e-3)** |
|  | $p$ | 8.650e-1 | 7.394e-1 | 7.394e-1 | - |
|  | $h$ | 0 | 0 | 0 | - |
| DF5 | (10,10) | 6.749e-3(4.252e-4) | **6.529e-3(6.999e-4)** | 6.650e-3(7.786e-4) | 6.832e-3(7.967e-4) |
|  | $p$ | 7.618e-1 | 4.643e-1 | 2.581e-1 | - |
|  | $h$ | 0 | 0 | 0 | - |
| DF6 | (10,10) | 6.252e-1(4.554e-2) | 5.974e-1(4.961e-2) | **5.763e-1(5.042e-2)** | 6.140e-1(4.412e-2) |
|  | $p$ | 9.941e-1 | 6.843e-1 | 5.997e-1 | - |
|  | $h$ | 0 | 0 | 0 | - |
| DF7 | (10,10) | 9.970e-3(2.978e-4) | 9.849e-3(3.574e-4) | 1.014e-2(4.601e-4) | **9.785e-3(4.370e-4)** |
|  | $p$ | 7.958e-1 | 8.543e-1 | 4.918e-1 | - |
|  | $h$ | 0 | 0 | 0 | - |
| DF8 | (10,10) | 7.178e-3(4.119e-4) | 7.262e-3(3.803e-4) | 7.296e-3(2.870e-4) | **7.152e-3(5.012e-4)** |
|  | $p$ | 8.650e-1 | 8.650e-1 | 7.172e-1 | - |
|  | $h$ | 0 | 0 | 0 | - |
| DF9 | (10,10) | 6.904e-2(5.571e-3) | **6.425e-2(5.298e-3)** | 6.444e-2(5.086e-3) | 6.755e-2(5.406e-3) |
|  | $p$ | 7.618e-1 | 5.592e-1 | 5.395e-1 | - |
|  | $h$ | 0 | 0 | 0 | - |
| DF10 | (10,10) | 1.048e-1(4.048e-3) | 1.021e-1(4.865e-3) | 9.985e-2(3.487e-3) | **1.001e-1(3.921e-3)** |
|  | $p$ | 3.020e-11 | 3.020e-11 | 3.020e-11 | - |
|  | $h$ | 1 | 1 | 1 | - |
| DF11 | (10,10) | 6.446e-1(1.322e-3) | 6.437e-1(1.628e-3) | 6.437e-1(1.571e-3) | **6.436e-1(1.138e-3)** |
|  | $p$ | 4.504e-11 | 4.504e-11 | 5.494e-11 | - |
|  | $h$ | 1 | 1 | 1 | - |
| DF12 | (10,10) | 2.841e-1(6.965e-3) | 2.841e-1(6.633e-3) | 2.851e-1(7.355e-3) | **2.831e-1(1.074e-2)** |
|  | $p$ | 3.338e-11 | 3.338e-11 | 3.338e-11 | - |
|  | $h$ | 1 | 1 | 1 | - |
| DF13 | (10,10) | 1.562e-1(6.031e-3) | **1.458e-1(4.852e-3)** | 1.461e-1(4.106e-3) | 1.500e-1(4.638e-3) |
|  | $p$ | 4.118e-6 | 2.278e-5 | 3.831e-5 | - |
|  | $h$ | 1 | 1 | 1 | - |
| DF14 | (10,10) | 5.336e-2(2.175e-3) | 5.277e-2(1.527e-3) | 5.263e-2(1.828e-3) | **5.251e-2(1.619e-3)** |
|  | $p$ | 4.504e-11 | 4.505e-11 | 4.504e-11 | - |
|  | $h$ | 1 | 1 | 1 | - |
|  | ‡/†/ℓ | 5/0/9 | 4/1/9 | 3/2/9 | - |

**Table 15. Performance comparison of FBP variants on MIGD for $(n_t, \tau_t) = (10,10)$.**

| Fun. | $(n_t, \tau_t)$ | FBPL1 | FBPL2 | FBPL3 | FBP |
|---|---|---|---|---|---|
| DF1 | (10,10) | **5.802e-3(2.648e-4)** | 5.817e-3(2.360e-4) | 5.878e-3(2.505e-4) | 5.844e-3(2.682e-4) |
| | $p$ | 7.958e-1 | 9.234e-1 | 8.534e-1 | - |
| | $h$ | 0 | 0 | 0 | - |
| DF2 | (10,10) | 4.474e-2(4.117e-3) | 4.165e-2(4.243e-3) | 4.435e-2(4.543e-3) | **3.937e-2(4.765e-3)** |
| | $p$ | 5.793e-1 | 9.117e-1 | 5.793e-1 | - |
| | $h$ | 0 | 0 | 0 | - |
| DF3 | (10,10) | 1.199e-2(6.526e-3) | 1.518e-2(1.194e-2) | 1.319e-2(8.732e-3) | **1.162e-2(5.596e-3)** |
| | $p$ | 9.587e-1 | 9.234e-1 | 8.073e-1 | - |
| | $h$ | 0 | 0 | 0 | - |
| DF4 | (10,10) | 7.126e-2(2.506e-3) | 7.098e-2(2.851e-3) | 7.043e-2(2.710e-3) | **7.029e-2(2.230e-3)** |
| | $p$ | 8.534e-1 | 8.650e-1 | 9.587e-1 | - |
| | $h$ | 0 | 0 | 0 | - |
| DF5 | (10,10) | 6.870e-3(8.297e-4) | **6.704e-3(7.194e-4)** | 6.736e-3(7.136e-4) | 6.832e-3(7.967e-4) |
| | $p$ | 8.534e-1 | 9.117e-1 | 8.883e-1 | - |
| | $h$ | 0 | 0 | 0 | - |
| DF6 | (10,10) | 6.212e-1(4.999e-2) | 6.137e-1(4.644e-2) | **5.932e-1(4.410e-2)** | 6.140e-1(4.412e-2) |
| | $p$ | 8.418e-1 | 9.117e-1 | 8.766e-1 | - |
| | $h$ | 0 | 0 | 0 | - |
| DF7 | (10,10) | 9.825e-3(3.817e-4) | 9.804e-3(3.990e-4) | 9.803e-3(3.770e-4) | **9.785e-3(4.370e-4)** |
| | $p$ | 8.418e-1 | 9.234e-1 | 8.766e-1 | - |
| | $h$ | 0 | 0 | 0 | - |
| DF8 | (10,10) | 7.104e-3(5.527e-4) | 7.155e-3(4.678e-4) | **6.994e-3(3.990e-4)** | 7.152e-3(5.012e-4) |
| | $p$ | 7.958e-1 | 9.352e-1 | 5.106e-1 | - |
| | $h$ | 0 | 0 | 0 | - |
| DF9 | (10,10) | 6.728e-2(6.911e-3) | 6.655e-2(4.605e-3) | **6.479e-2(4.734e-3)** | 6.755e-2(5.406e-3) |
| | $p$ | 8.187e-1 | 8.650e-1 | 6.627e-1 | - |
| | $h$ | 0 | 0 | 0 | - |
| DF10 | (10,10) | 1.010e-1(5.205e-3) | 1.005e-1(3.651e-3) | **9.989e-2(5.214e-3)** | 1.001e-1(3.921e-3) |
| | $p$ | 3.020e-11 | 3.020e-11 | 3.020e-11 | - |
| | $h$ | 1 | 1 | 1 | - |
| DF11 | (10,10) | 6.465e-1(1.292e-3) | 6.441e-1(1.271e-3) | 6.475e-1(1.436e-3) | **6.436e-1(1.138e-3)** |
| | $p$ | 4.975e-11 | 4.975e-11 | 5.494e-11 | - |
| | $h$ | 1 | 1 | 1 | - |
| DF12 | (10,10) | 2.843e-1(7.265e-3) | 3.143e-1(1.567e-2) | 2.931e-1(1.315e-2) | **2.831e-1(1.074e-2)** |
| | $p$ | 3.338e-11 | 3.020e-11 | 3.020e-11 | - |
| | $h$ | 1 | 1 | 1 | - |
| DF13 | (10,10) | 1.503e-1(5.011e-3) | 1.517e-1(6.104e-3) | 1.587e-1(5.111e-3) | **1.500e-1(4.638e-3)** |
| | $p$ | 1.635e-5 | 1.337e-5 | 1.635e-5 | - |
| | $h$ | 1 | 1 | 1 | - |
| DF14 | (10,10) | 5.318e-2(2.072e-3) | 5.295e-2(1.852e-3) | 5.299e-2(2.060e-3) | **5.251e-2(1.619e-3)** |
| | $p$ | 4.975e-11 | 4.504e-11 | 4.504e-11 | - |
| | $h$ | 1 | 1 | 1 | - |
| | ‡/†/ℓ | 5/0/9 | 5/0/9 | 4/1/9 | - |

**Table 16. Performance comparison of FBP variants on MIGD for $(n_t, \tau_t)$ = (10,10).**

| Fun. | $(n_t, \tau_t)$ | FBPR1 | FBPR2 | FBPR3 | FBP |
|---|---|---|---|---|---|
| DF1 | (10,10) | 6.377e-3(2.337e-4) | 6.404e-3(2.941e-4) | 6.730e-3(3.582e-4) | **5.844e-3(2.682e-4)** |
| | $p$ | 2.324e-2 | 2.416e-2 | 4.033e-3 | - |
| | $h$ | 1 | 1 | 1 | - |
| DF2 | (10,10) | 3.503e-2(3.508e-3) | 3.478e-2(3.890e-3) | **3.397e-2(3.457e-3)** | 3.937e-2(4.765e-3) |
| | $p$ | 8.883e-1 | 9.234e-1 | 7.062e-1 | - |
| | $h$ | 0 | 0 | 0 | - |
| DF3 | (10,10) | 1.527e-2(7.933e-3) | 1.186e-2(4.311e-3) | 1.478e-2(7.774e-3) | **1.162e-2(5.596e-3)** |
| | $p$ | 6.669e-3 | 1.564e-2 | 4.217e-4 | - |
| | $h$ | 1 | 1 | 1 | - |
| DF4 | (10,10) | 7.054e-2(2.689e-3) | 7.350e-2(3.801e-3) | 7.238e-2(2.734e-3) | **7.029e-2(2.230e-3)** |
| | $p$ | 7.062e-1 | 6.952e-1 | 7.172e-1 | - |
| | $h$ | 0 | 0 | 0 | - |
| DF5 | (10,10) | 8.473e-3(6.716e-4) | 8.871e-3(9.179e-4) | 9.923e-3(1.029e-4) | **6.832e-3(7.967e-4)** |
| | $p$ | 2.499e-3 | 2.380e-3 | 1.784e-4 | - |
| | $h$ | 1 | 1 | 1 | - |
| DF6 | (10,10) | 7.340e-1(3.647e-2) | 7.413e-1(4.230e-2) | 7.495e-1(5.301e-2) | **6.140e-1(4.412e-2)** |
| | $p$ | 2.838e-1 | 2.838e-1 | 2.519e-1 | - |
| | $h$ | 0 | 0 | 0 | - |
| DF7 | (10,10) | 9.764e-3(5.243e-4) | 9.811e-3(4.268e-4) | **9.528e-3(3.826e-4)** | 9.785e-3(4.370e-4) |
| | $p$ | 8.073e-1 | 8.073e-1 | 5.592e-1 | - |
| | $h$ | 0 | 0 | 0 | - |
| DF8 | (10,10) | 8.419e-3(6.807e-4) | 8.155e-3(5.792e-4) | 8.814e-3(7.317e-4) | **7.152e-3(5.012e-4)** |
| | $p$ | 1.628e-2 | 4.207e-2 | 3.183e-3 | - |
| | $h$ | 1 | 1 | 1 | - |
| DF9 | (10,10) | 6.938e-2(4.694e-3) | 7.013e-2(5.815e-3) | 7.453e-2(6.473e-3) | **6.755e-2(5.406e-3)** |
| | $p$ | 8.534e-1 | 6.627e-1 | 3.555e-1 | - |
| | $h$ | 0 | 0 | 0 | - |
| DF10 | (10,10) | 1.039e-1(4.889e-3) | 1.079e-1(6.567e-3) | 1.113e-1(5.343e-3) | **1.001e-1(3.921e-3)** |
| | $p$ | 3.020e-11 | 3.020e-11 | 3.020e-11 | - |
| | $h$ | 1 | 1 | 1 | - |
| DF11 | (10,10) | 6.442e-1(1.673e-3) | 6.442e-1(1.033e-3) | 6.448e-1(1.783e-3) | **6.436e-1(1.138e-3)** |
| | $p$ | 4.975e-11 | 5.494e-11 | 4.504e-11 | - |
| | $h$ | 1 | 1 | 1 | - |
| DF12 | (10,10) | 2.942e-1(8.063e-3) | 2.958e-1(7.183e-3) | 2.951e-1(9.245e-3) | **2.831e-1(1.074e-2)** |
| | $p$ | 3.338e-11 | 3.338e-11 | 3.338e-11 | - |
| | $h$ | 1 | 1 | 1 | - |
| DF13 | (10,10) | 1.547e-1(6.811e-3) | 1.594e-1(4.87e-3) | 1.576e-1(7.544e-3) | **1.500e-1(4.638e-3)** |
| | $p$ | 6.283e-6 | 2.491e-6 | 2.879e-6 | - |
| | $h$ | 1 | 1 | 1 | - |
| DF14 | (10,10) | 5.271e-2(1.840e-3) | 5.316e-2(2.267e-3) | 5.367e-2(2.809e-3) | **5.251e-2(1.619e-3)** |
| | $p$ | 4.504e-11 | 4.504e-11 | 4.504e-11 | - |
| | $h$ | 1 | 1 | 1 | - |
| | ‡/†/≀ | 9/0/5 | 9/0/5 | 9/0/5 | - |

all the problems. Especially in some cases, the difference between them are quite significant i.e., DF1, DF5, DF10. Therefore, 0.1 is the best one for *cr* in this study.

## 5.5 Influence of *Subpop$_2$* size

Another important parameter is the *Subpop$_2$* size. To investigate its influence, the *Subpop$_2$* size changes with an increment of 0.1 from 0.2 to 0.5 times of the total population size (FBPQ1-FBPQ3). The comparison results recorded in Table 17 show that there is no best values for this parameter for all the test functions. For instance, some cases (e.g. DF3 and DF13) are sensitive to the parameter value, while other cases (e.g. DF1 and DF2) are not affected by this parameter too much. This experiment supports that FBP has much better performance compared with the other variants when *Subpop$_2$* is defined as around $0.4N$, although it is not always the best. Thus, $0.4N$ is chosen for this parameter in FBP.

## 5.6 Different Multi-objective algorithms

This subsection aims to verify the feasibility of the proposed dynamic reaction mechanism by combining it with four efficient and new proposed multiobjective algorithms.

## 5.7 More discussion

Apart from the aforementioned component and parameter analysis, this subsection further discusses the advantages and disadvantages of each strategy of the proposed technique. Firstly, the linear prediction strategy utilizes the two-step strategy for predicting non-dominated solutions, which increases the quality of the population in dynamic environments and improves the optimization performance. However, improvement comes at the cost of complexity, since compared with one-step strategy, the two-step strategy tends to generate more solutions. Meanwhile, these solutions contain some boundary individuals, which are not beneficial for global search, as shown in the numerical results where these boundary individuals are non-dominated. Therefore, this strategy should be modified by controlling the boundary members effectively.

Secondly, to obtain well-distributed solutions, FBP employs a recent sampling strategy by classifying decision variables into two groups. Experimental results also show that it is also an effective way for solving multiobjective problems in varying environments. However, the strategy heavily depends on variable classification. This study assumes that there exists principle and non-principle variables, but it not clear about the generalisation of this assumption. Thus, this strategy also needs to be improved effectively to avoid the principal being misidentified.

Thirdly, the curve-fitting based strategy aims to predict a subpopulation based on the distribution characteristic among variables in two consecutive environments. Simulation results show that it enhances performance in bi-objective problems, but is not helpful for triple-objective problems. Therefore, further improvement should be make on this strategy.

## 6 Conclusion

This paper proposed a new dynamic multiobjective optimization algorithm, named FBP, for dealing with multiobjective problems in changing environments. FBP mainly includes three different components, that is, a two-step approach for predicting non-dominated solutions, a sampling strategy and a curve-fitting strategy. Each component has an important role for create high-quality population, improving either diversity or convergence, when a change occurs in the environment. To verify the effectiveness of our algorithm, a recent test suite with different characteristics is utilized. Experimental comparisons demonstrate that FBP has better

**Table 17. Performance comparison of FBP variants on MIGD for $(n_t, \tau_t) = (10,10)$.**

| Fun. | $(n_t, \tau_t)$ | FBPQ1 | FBPQ2 | FBPQ3 | FBP |
|------|------|-------|-------|-------|-----|
| DF1 | (10,10) | 5.911e-3(3.821e-4) | 5.858e-3(1.873e-4) | 5.948e-3(3.004e-4) | **5.844e-3(2.682e-4)** |
| | $p$ | 9.705e-1 | 9.234e-1 | 8.187e-1 | - |
| | $h$ | 0 | 0 | 0 | - |
| DF2 | (10,10) | 3.973e-2(4.259e-3) | **3.852e-2(3.999e-3)** | 3.938e-2(5.429e-3) | 3.937e-2(4.765e-3) |
| | $p$ | 9.000e-1 | 9.941e-1 | 9.117e-1 | - |
| | $h$ | 0 | 0 | 0 | - |
| DF3 | (10,10) | 1.352e-2(1.137e-2) | **1.214e-2(8.241e-3)** | 1.494e-2(7.493e-3) | 1.162e-2(5.596e-3) |
| | $p$ | 9.823e-1 | 9.352e-1 | 9.352e-1 | - |
| | $h$ | 0 | 0 | 0 | - |
| DF4 | (10,10) | 7.070e-2(2.6506e-3) | 7.076e-2(2.721e-3) | 7.118e-2(3.231e-3) | **7.029e-2(2.230e-3)** |
| | $p$ | 9.000e-1 | 9.117e-1 | 9.941e-1 | - |
| | $h$ | 0 | 0 | 0 | - |
| DF5 | (10,10) | 7.061e-3(9.695e-4) | 6.897e-3(7.156e-4) | **6.744e-3(5.833e-4)** | 6.832e-3(7.967e-4) |
| | $p$ | 9.470e-1 | 9.823e-1 | 9.352e-1 | - |
| | $h$ | 0 | 0 | 0 | - |
| DF6 | (10,10) | 6.650e-1(4.967e-2) | 6.261e-1(4.250e-2) | **5.806e-1(3.250e-2)** | 6.140e-1(4.412e-2) |
| | $p$ | 7.172e-1 | 8.883e-1 | 6.843e-1 | - |
| | $h$ | 0 | 0 | 0 | - |
| DF7 | (10,10) | **9.732e-3(3.982e-4)** | 9.779e-3(3.525e-4) | 9.744e-3(3.357e-4) | 9.785e-3(4.370e-4) |
| | $p$ | 9.352e-1 | 9.470e-1 | 8.766e-1 | - |
| | $h$ | 0 | 0 | 0 | - |
| DF8 | (10,10) | 7.104e-3(3.891e-4) | **7.085e-3(3.684e-4)** | 7.092e-3(3.835e-4) | 7.152e-3(5.012e-4) |
| | $p$ | 9.823e-1 | 8.650e-1 | 9.823e-1 | - |
| | $h$ | 0 | 0 | 0 | - |
| DF9 | (10,10) | 6.762e-2(5.423e-3) | 6.641e-2(5.793e-3) | **6.461e-2(4.087e-3)** | 6.755e-2(5.406e-3) |
| | $p$ | 9.352e-1 | 8.073e-1 | 6.735e-1 | - |
| | $h$ | 0 | 0 | 0 | - |
| DF10 | (10,10) | 1.024e-1(5.649e-3) | 1.013e-1(4.576e-3) | 1.024e-1(6.013e-3) | **1.001e-1(3.921e-3)** |
| | $p$ | 3.020e-11 | 3.020e-11 | 3.020e-11 | - |
| | $h$ | 1 | 1 | 1 | - |
| DF11 | (10,10) | 6.442e-1(1.810e-3) | 6.445e-1(1.437e-3) | 6.439e-1(1.635e-3) | **6.436e-1(1.138e-3)** |
| | $p$ | 4.504e-11 | 4.975e-11 | 6.066e-11 | - |
| | $h$ | 1 | 1 | 1 | - |
| DF12 | (10,10) | **2.793e-1(4.853e-3)** | 2.795e-1(7.997e-3) | 2.887e-1(7.027e-3) | 2.831e-1(1.074e-2) |
| | $p$ | 3.338e-11 | 3.338e-11 | 3.338e-11 | - |
| | $h$ | 1 | 1 | 1 | - |
| DF13 | (10,10) | 1.521e-1(6.115e-3) | 1.516e-1(5.229e-3) | 1.518e-1(5.559e-3) | **1.500e-1(4.638e-3)** |
| | $p$ | 9.514e-6 | 1.249e-5 | 1.091e-5 | - |
| | $h$ | 1 | 1 | 1 | - |
| DF14 | (10,10) | 5.269e-2(1.620e-3) | 5.328e-2(1.980e-3) | 5.293e-2(1.755e-3) | **5.251e-2(1.619e-3)** |
| | $p$ | 4.504e-11 | 4.504e-11 | 4.504e-11 | - |
| | $h$ | 1 | 1 | 1 | - |
| | ‡/†/≀ | 4/1/9 | 4/1/9 | 5/0/9 | - |

performance than the other algorithms on most cases, showing the proposed algorithm has a good tracking ability and responds fast to environmental changes. Besides, the role that each component and parameter plays in the proposed algorithm is also analysed and discussed extensively. In our future work, we will further improve the proposed algorithm by addressing some parameter issues as discussed previously.

## Supporting information

**S1 File.**
(ZIP)

## Acknowledgments

The authors express sincerely appreciation to the anonymous reviewers for their helpful opinions.

## Author Contributions

**Investigation:** Qingyang Zhang.

**Methodology:** Qingyang Zhang, Shouyong Jiang, Shengxiang Yang, Hui Song.

**Resources:** Shengxiang Yang.

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
