## [Decision Letter · Decision Letter 0]

4 May 2021

PONE-D-21-11897

Solving dynamic multi-objective problems with a new prediction-based optimization algorithm

PLOS ONE

We look forward to receiving your revised manuscript.

Kind regards,

Seyedali Mirjalili

Academic Editor

PLOS ONE

Journal Requirements:

2.We suggest you thoroughly copyedit your manuscript for language usage, spelling, and grammar. If you do not know anyone who can help you do this, you may wish to consider employing a professional scientific editing service.  

Reviewers' comments:

Reviewer's Responses to Questions

**Comments to the Author**

1. Is the manuscript technically sound, and do the data support the conclusions?

Reviewer #1: Yes

2. Has the statistical analysis been performed appropriately and rigorously? 

Reviewer #1: No

3. Have the authors made all data underlying the findings in their manuscript fully available?

Reviewer #1: Yes

4. Is the manuscript presented in an intelligible fashion and written in standard English?

Reviewer #1: Yes

5. Review Comments to the Author

Reviewer #1: The authors proposed a dynamic multi-objective optimization algorithm by integrating a new fitting based prediction (FBP) mechanism, good work; however, there are major comments to be addressed as follows:

1) Introduction part is somehow poor, authors need to include a general description of Multi-objective optimization and the need for the multi-objective idea, then come to the dynamic multi-objective.

2) The contribution should be clearly explained and this usually comes from the literature gaps.

3) There is no section for related work. It is recommended to include a section named Related Work, in this section, the authors should critically review the recent studies on MO and DMO and find the limitations and drawbacks that drawn the problem they are trying to solve.

4) Recent MO algorithms could be used for comparisons such as MOGWO, MOWOA, MOPSO, and others.

5) No statistical test performed to find the significant difference between the proposed method and the state-of-the-art. It is recommended to use Anova test or t-test to find the p-value at a significant level less than 0.05.

6. PLOS authors have the option to publish the peer review history of their article (what does this mean?). If published, this will include your full peer review and any attached files.

Reviewer #1: No

---

## [Author Response · Author response to Decision Letter 0]

24 Jun 2021

Revision Statement for "Solving dynamic multi-objective problems with a new prediction-based optimization algorithm"( No.PONE-D-21-11897)

The referees' reviews are greatly appreciated and carefully considered in the revised version of the manuscript. The authors' responses to the reviewers and main changes in the revised version are summarized as follows.

Academic Editor

Comment: Please submit your revised manuscript by Jun 18 2021 11:59PM.

Response: Many thanks for the AE's time and assistance in dealing with our manuscript. We have carefully considered the reviewers' comments in the revision, and detailed responses to the comments are given as follows.

Reviewer 1

Comment 1: Introduction part is somehow poor, authors need to include a general description of Multi-objective optimization and the need for the multi-objective idea, then come to the dynamic multi-objective.

Response: Many Thanks for the reviewer's valuable comments. We have revised the introduction section by a more rigorous literature review of the proposed area. The relevant description and concept of Multi-objective optimization have been added and marked in Introduction section in the 'Revised Manuscript with Track Changes'.

Comment 2: The contribution should be clearly explained and this usually comes from the literature gaps.

Response: We totally agree with the reviewer on this point. Therefore, we have cited the relevant literature gaps and added the relevant contents in the revised paper.

"...they neglect properties of decision variables, which is an important part of discovering.... Simultaneously, according to [54-55], curve fitting technique is a classic and popular technique, ... relationship between variables to a certain extent and predict possible regions or directions..."

Comment 3: There is no section for related work. It is recommended to include a section named Related Work, in this section, the authors should critically review the recent studies on MO and DMO and find the limitations and drawbacks that drawn the problem they are trying to solve.

Response: Many thanks for your suggestion. In the revised paper, we have added the related work as Section 2 by increasing the review of on MO and DMO and providing their weaknesses. In detailed, the review of MO algorithms are divided into three categories, and some representative algorithms and limitations are also listed. For DMO, the existing algorithms were divided into four categories, and some representative algorithms and drawbacks are also presented.

"...As one of the most attractive and popular areas in intelligent computing field, ... optimization algorithms can be classified into three categories as follows...These algorithms can obtain good local optimal solutions, ...but it is difficult to achieve ideal global optimal solutions...Existing Dynamic multi-objective optimization algorithms can be divided into four categories: ..., and prediction-based algorithms...."

Comment 4: Recent MO algorithms could be used for comparisons such as MOGWO, MOWOA, MOPSO, and others.

Response: We have enhanced experimental studies by carrying out more simulations according to this comment. We have compared with some well and recent MO algorithms (MOGOA, MOMVO, MOALO and MSSA) to show the effectiveness of the proposed algorithm. In the revised manuscript, the relevant contents can be found in Section 4, and the simulation and comparison results are recorded in Table 10-12.

Comment 5: No statistical test performed to find the significant difference between the proposed method and the state-of-the-art. It is recommended to use Anova test or t-test to find the p-value at a significant level less than 0.05.

Response: Many thanks for pointing this out. We have added the t-test indicator in experimental studies, and summarized the comparison results at the bottom of each Table. This content can be found in all Tables in the revised manuscript.

Many thanks for the supportive comments. We hope our effort address the reviewer's concerns with satisfaction. 

Kind Regards

---

## [Decision Letter · Decision Letter 1]

5 Jul 2021

Solving dynamic multi-objective problems with a new prediction-based optimization algorithm

PONE-D-21-11897R1

Dear Dr. zhang,

We’re pleased to inform you that your manuscript has been judged scientifically suitable for publication and will be formally accepted for publication once it meets all outstanding technical requirements.

Kind regards,

Seyedali Mirjalili

Academic Editor

PLOS ONE

Additional Editor Comments (optional):

Reviewers' comments:

Reviewer's Responses to Questions

**Comments to the Author**

1. If the authors have adequately addressed your comments raised in a previous round of review and you feel that this manuscript is now acceptable for publication, you may indicate that here to bypass the “Comments to the Author” section, enter your conflict of interest statement in the “Confidential to Editor” section, and submit your "Accept" recommendation.

Reviewer #1: All comments have been addressed

2. Is the manuscript technically sound, and do the data support the conclusions?

Reviewer #1: Yes

3. Has the statistical analysis been performed appropriately and rigorously? 

Reviewer #1: Yes

4. Have the authors made all data underlying the findings in their manuscript fully available?

Reviewer #1: Yes

5. Is the manuscript presented in an intelligible fashion and written in standard English?

Reviewer #1: Yes

6. Review Comments to the Author

Reviewer #1: The authors did great efforts in response to the comments given in the first round of review. Most of the comments have been addressed. Therefore, the paper is acceptable for publication.

7. PLOS authors have the option to publish the peer review history of their article (what does this mean?). If published, this will include your full peer review and any attached files.

Reviewer #1: No

---

## [Editor Report · Acceptance letter]

21 Jul 2021

PONE-D-21-11897R1 

Solving dynamic multi-objective problems with a new prediction-based optimization algorithm 

Dear Dr. Zhang:

I'm pleased to inform you that your manuscript has been deemed suitable for publication in PLOS ONE. Congratulations! Your manuscript is now with our production department. 

Kind regards, 

on behalf of

Prof. Seyedali Mirjalili 

Academic Editor

PLOS ONE